# Organic Farming Lessens Reliance on Pesticides and Promotes Public Health by Lowering Dietary Risks

**Charles Benbrook** [1,*], **Susan Kegley** [2] **and Brian Baker** [3]

[1] Heartland Health Research Alliance, Port Orchard, WA 98367, USA

[2] Pesticide Research Institute, Santa Rosa, CA 95404, USA; skegley@pesticideresearch.com

[3] Crop and Soil Science Department, Oregon State University, Corvallis, OR 97331, USA; bakebria@oregonstate.edu

*   Correspondence: cbenbrook@hh-ra.org

**Abstract:** Organic agriculture is a production system that relies on prevention, ecological processes, biodiversity, mechanical processes, and natural cycles to control pests and maintain productivity. Pesticide use is generally limited or absent in organic agroecosystems, in contrast with non-organic (conventional) production systems that primarily rely on pesticides for crop protection. Significant differences in pesticide use between the two production systems markedly alter the relative dietary exposure and risk levels and the environmental impacts of pesticides. Data are presented on pesticide use on organic and non-organic farms for all crops and selected horticultural crops. The relative dietary risks that are posed by organic and non-organic food, with a focus on fresh produce, are also presented and compared. The results support the notion that organic farms apply pesticides far less intensively than conventional farms, in part because, over time on well-managed organic farms, pest pressure falls when compared to the levels on nearby conventional farms growing the same crops. Biopesticides are the predominant pesticides used in organic production, which work by a non-toxic mode of action, and pose minimal risks to human health and the environment. Consequently, eating organic food, especially fruits and vegetables, can largely eliminate the risks posed by pesticide dietary exposure. We recommend ways to lower the pesticide risks by increased adoption of organic farming practices and highlight options along organic food supply chains to further reduce pesticide use, exposures, and adverse worker and environmental impacts.

**Keywords:** organic farming; pesticides; public health; pesticide use; biopesticides; integrated pest management; food quality protection

## 1. Introduction

Organic farming systems have long been compared with non-organic (conventional) farming systems in numerous scientific studies and official reports [1–4]. One significant difference is the way that pest management tools, tactics, practices, and inputs are integrated into systems to prevent economically damaging pest losses [5–7].

Organic growers use ecological principles and practices to build and sustain diverse communities of below- and above-ground organisms in ways that generally prevent economic losses from arthropod pests, plant pathogens, and weeds [8,9]. Combinations of cultural, physical, and mechanical practices are utilized to control weeds on organic farms, and herbicides are rarely used [10]. In contrast, conventional farmers generally rely predominantly and often exclusively on pesticides to manage pests, diseases, and weeds [5,11].

### 1.1. Differences between Conventional and Organic Farming

Organic farmers tend to have a higher threshold for pest pressure and weed competition than the typical conventional farmer. In general, pest-driven losses in crop yield and

quality are greater on organic farms than nearby conventional farms, and they are also more variable. Pest losses and lower yields increase the production costs per unit and depress profits on organic farms, but reductions in the operating costs and higher prices increase profits [12].

One of the few detailed comparisons of production costs and profits on organic versus conventional farms producing the same commodity focused on corn production in the US in 2010 [13]. This USDA analysis reported 81% higher net returns on the organic farms ($1371/ Ha or $555/A vs. $759/Ha or $307/A), 39% lower seed+fertilizer+chemical costs ($544/Ha or $220/A conventional vs. $331/Ha or $134/A organic), and 22% lower total operating costs ($717/Ha or $290/A conventional vs. $565/Ha or $229/A organic). Yields on the conventional farms exceeded organic yields by 31%, but the price was 72% higher for the organic corn ($12.29 conventional vs. $21.17 organic). The gross value of production was $2233 per hectare ($904 per acre) on the organic corn farms and $1702 per hectare ($689 per acre) on the conventional farms.

Most registered pesticides are prohibited for use in organic production. In the United States, the US Department of Agriculture (USDA) established the list of pesticides and other agricultural inputs that may be used on organic farms under the statutory authority of the Organic Foods Production Act [14] and the regulatory authority of the National Organic Program (NOP) [15]. The NOP establishes a list of synthetic substances that may be used on organic farms under certain circumstances (hereafter, the "National list"). Pesticides that meet strict criteria are added to the National list with consideration of: (a) impacts on human health and the environment and (b) the source of active ingredients and the risks that are posed by inert ingredients included in end-use product formulations.

Non-synthetic (natural) substances may be used as pesticides unless explicitly prohibited because of significant risks to human health and/or the environment (e.g., arsenic, nicotine, rotenone, strychnine, lead, and mined cryolite [sodium fluoroaluminate]) [16].

Most of the pesticides that are approved for use on organic farms fall within the EPA's definition of "biopesticide". The Environmental Protection Agency defines biopesticides as "naturally occurring substances that control pests (biochemical pesticides), microorganisms that control pests (microbial pesticides), and pesticidal substances that are produced by plants containing added genetic material (plant-incorporated protectants) or PIPs" [17]. Most, but not all, biopesticides work through a non-toxic mode of action.

Biopesticides that are approved for use on organic farms include copper, horticultural oils, kaolin clay, sulfur, and lime-sulfur compounds; toxins that are derived from bacteria (e.g., many varieties of *Bacillus thuringinesis* endotoxins [*Bt*]); botanical extracts; insect traps and sticky barriers; insect pheromones; and soaps [18]. These appear on the National List following a public petition, technical review, and public consultation. The list is reviewed every five years. When new facts indicate that certain substances are not consistent with the criteria set out in the NOP rules, the substances may be removed (e.g., the antibiotic-based fungicides streptomycin and tetracycline were removed from the National List in 2017).

In addition, regulatory authorities have classified most biopesticides as exempt from the requirement of a tolerance because they pose low or very low dietary risk. Currently, 44 substances are also exempt from the requirement of registration based on the minimal risk that they pose to human health and the environment [19].

The US EPA has exempted all but one pesticide used for organic production from the requirement of a food tolerance. The EPA grants such exemptions for pesticide uses that lead to no or minimal dietary risk. Spinosad is the one exception, which is a widely used insecticide on both organic and conventional fruit and vegetable farms. Spinosad is composed of a complex of toxins that are derived from a naturally occurring soil microorganism. Residues in food are common, and they are typically reported as Spinosad A and Spinosad D. The manufacturer (Corteva/Dow) offers two sets of formulations: one that is

marketed for use on conventional farms (Success) and a second (Entrust) with co-formulants or "inert" ingredients that are acceptable to the NOP and organic certifiers [20,21].

NOP regulations limit use of pesticides to only when biological, cultural, and physical practices are ineffective [22]. Across all pests and regions when designing and deploying pest management systems, organic farmers typically primarily depend on management options and non-synthetic chemical control tactics, while conventional farmers primarily depend on pesticides [11].

Organic farmers are required to prepare an Organic System Plan (OSP) that describes the cultural, mechanical, and biological practices to be undertaken, the monitoring system to determine whether such practices are effective, and the specific pesticides to be used if preventive practices are ineffective [23]. OSPs also typically specify the pest population thresholds or other field-based diagnostic criteria that will be tracked and relied on to justify a chemical intervention. A USDA Accredited Certification Agent (ACA) must review and approve the plan. When pesticides are used in organic farming systems, they are subject to additional limitations beyond what is required on pesticide product labels.

Most co-formulants of organic pesticide products are selected from a list of inert ingredients that are deemed by the EPA to be of minimal concern [24]. Specific brand name products, including their co-formulants, must be reviewed and approved by an ACA and/or an USDA accredited materials review organization, and included in a farm's OSP before an organic farmer can use them.

In the US under the NOP, organic farms are inspected every year to ensure compliance with NOP rules [25]. ACAs must collect samples and test them for prohibited substances in foods annually from 5% of the organic farms that they certify. Such sampling can be either random or for cause—e.g., a report of a possible illegal application, a split operation that produces and handles a large volume of organic and non-organic product, or an operation with a history of non-compliance [26]. Organic food that tests positive for a pesticide may not be sold as organic if the residue detected is 5% or more of the applicable EPA tolerance for that pesticide-food combination [27].

On established, well-managed organic farms, pest management systems deliver clear advantages when compared to well-managed neighboring conventional farms. Organically managed agroecosystems contribute to:

- increased organic matter and improved carbon sequestration, coupled with enhanced soil health [3,28–31];
- restored biodiversity [7,32–35];
- improved pollinator performance [36,37];
- consumer benefits via less frequent and lower risk residues in food [38–40];
- reduced broad-spectrum pesticide applications that sometimes trigger secondary pest outbreaks [5,11,41]; and
- increased productivity and resilience [42,43];
- improved water quality [44,45].

Despite desirable benefits, organic production systems account for under 10% of the national acreage of most crops in the US and less than 2% of all crops (ERS and NASS data). Several factors slow the transition to organic systems in the US:

- weed management challenges [46];
- cost of organic certification [46,47];
- recordkeeping requirements of organic certification [46,47];
- lack of financial support during transition, when farmers face lower yields, higher costs, and generally receive conventional prices [48,49];
- insufficient premium prices to cover higher costs of production after being certified organic [48];
- concern over near-term yield and crop quality losses [12];
- challenges inherent in learning a markedly different paradigm for crop production and pest management with little or no technical support from existing institutions [50];

- fraud that undercuts the organic premium and undermines confidence in the organic label [51]; and
- lack of sufficient infrastructure [50].

Pest management challenges become progressively less common, more easily isolated and overcome on most well-managed organic farms [42,52]. New challenges periodically arise, and they require system changes and innovation to restore acceptable levels of control, but adaptive and pro-active management can usually overcome these challenges without resorting to heavy pesticide use or reverting to conventional production. However, total crop losses on certain blocks on organic fruit and vegetable farms occasionally occur (see Figure 1 for a photo of an organic tomato field in Florida hit by late blight just days before harvest).

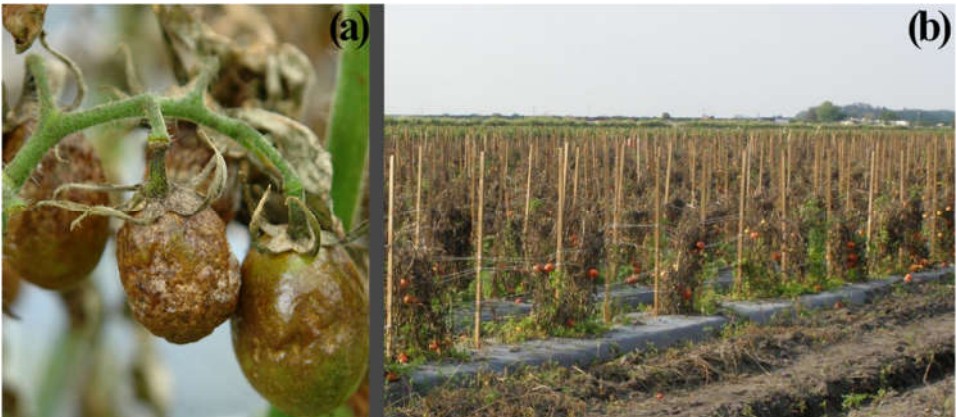

**Figure 1.** (**a**) Late blight is a disease of tomatoes and potatoes caused by the pathogen *Phytophthora infestans* that strikes late in the season and can ruin a crop close to harvest time. Photo: Wikimedia Commons; (**b**) This 20-acre block of organic tomatoes in Florida, valued at over half a million US dollars, was ruined by late blight in 2006 just days before harvest. Despite this loss, this diversified operation generated strong profits from the approximately 20 other crops grown on the farm. Photo: Charlie Mellinger.

Nearby, conventional tomato growers applied four to six fungicide applications that season to combat early and late blight and avoided significant crop loss.

### 1.2. Reduction in Pesticide Risk Drives Consumer Choices

Consumer surveys have shown that the desire to reduce pesticide dietary exposures is a primary reason people switch to organic foods [53,54]. The "Dirty Dozen" and "Clean 15" lists of foods that were issued by the Environmental Working Group [55] have raised awareness of the presence of residues in various foods and helped consumers to understand that:

- some common fruits and vegetables rarely contain pesticide residues, while other fresh produce items contain four or more residues, and a few have 10 or more;
- residue profiles and risks often differ substantially between domestically grown and imported foods; and
- organically grown food offers the surest way to markedly reduce pesticide dietary exposures and risks.

Biomedical research, toxicology, and epidemiology are making progress in identifying how pesticide exposures contribute to adverse health outcomes. Four health concerns warrant more systematic and focused research, and they are likely to further motivate consumers to seek out organic brands:

- Impacts on human reproduction and children's development, especially of the nervous, immune, and reproductive systems [56–63].

- Pesticide exposures and cancer, especially blood and brain cancers in children [64–68].
- Ways that chronic pesticide dietary exposure may be altering the composition and function of an individual's GI-tract microbiome [69,70].
- Whether and to what extent prenatal pesticide exposures are triggering heritable, epigenetic changes that increase the risk of adult-onset disease, reproductive abnormalities, or other health problems [71–75].

*1.3. Comparison of Pesticide Use and Risk in Organic and Conventional Systems*

A holistic, comparative assessment of pesticide use in organic versus conventional agricultural, food, and farming systems should account for the differences in four categories of pest-management system impacts:

1. The number, volume, and toxicity of pesticides applied.
2. Risks that are caused by exposure to residues in drinking water, beverages, and food.
3. Impacts on non-target organisms, such as farmworkers, pollinators, and earthworms.
4. Environmental and ecological impacts with both short-term (surface water runoff, spread of resistant weeds, and secondary pest outbreaks) and long-run consequences (contaminated aquifers, impaired soil health, and loss of biodiversity).

This paper presents data and primarily focuses on the first two categories of impacts. Differences in pesticide use correlate with changes in impacts on non-target organisms and environmental impacts, but a comprehensive treatment of categories 3 and 4 is beyond the scope of a single paper.

For conventional, integrated, and organic farming systems, the number of pesticides applied, the area treated, and treatment rates are higher on horticultural crops than for arable and forage crops. There are 50 to 100 active ingredients that are registered for use on most conventional crops; these active ingredients are sold via 1000 or more end-use products. The active ingredients are blended with co-formulants (inert ingredients) to make end-use pesticide products. Markedly fewer active ingredients and formulated products are approved for use on organic farms.

The co-formulants in pesticide products often alter the physical and chemical properties of the end-use product, in contrast to pure active ingredients [73,74]. For some of the most widely used pesticides, co-formulants markedly increase the innate toxicity of the formulated product, while also altering the pesticide's Absorption, Distribution, Metabolism, and Excretion (ADME). This is especially the case with many widely used, post-emergence herbicides that contain co-formulants that are chosen to enhance adherence to, and movement through, the epidermis of weeds. Such co-formulants essentially do the same thing when spray mist lands on human skin [76–78]. Glyphosate-based herbicides [78–80], neonicotinoid insecticides [78,81,82], and several important fungicides [78] are important examples of formulated products that pose greater risks than the pure active ingredients in them.

In addition, co-formulants in end-use products that are applied on conventional farms are sometimes present at concentrations approaching or even exceeding the concentration of active ingredient (e.g., Lorsban Advanced [83]). Some are known, or are possible human carcinogens, and they can increase the diversity of health risks. Others enhance product volatility, thereby altering worker exposure and risk profiles and the propensity to drift off target.

Residues of pesticides that are prohibited for organic production are sometimes detected in organic foods. Most can be traced to the following sources:

- persistent, legacy insecticides like DDT, aldrin-dieldrin, chlordane, and other organochlorine products not applied since the 1970s [84];
- post-harvest fungicides used in packing plants processing both conventional and organic produce, especially fruits [85];
- synthetic pesticides that drift or otherwise move from conventionally managed fields onto organic crops growing nearby; and

- deliberate, fraudulent applications, mislabeling, or negligent handling along supply chains.

The examples and data in this paper primarily reflect organic and conventional farming systems in the US, but the general findings and implications that arise from a decision to transition a conventionally managed operation to organic are comparable around the world. After summarizing the differences in pesticide use on selected organic and conventional farms, and comparing residue and risk profiles, we address R&D challenges, policy interventions, and organic certification issues. We do so with a focus on assuring that certified organic food delivers on its promise to lessen the public health, farmworker, and environmental risks that arise from the annual and inevitable challenges that all farmers face in managing pests.

## 2. Materials and Methods

The study relies on official sources of data from the USDA and California Department of Pesticide Regulation (CDPR) to analyze pesticide use, pesticide residues, and risks that arise from dietary exposure to pesticides. The residue data are analyzed based on the US EPA's dietary risk assessment methodologies [86]. Additional observations to help interpret the results are made in the discussion section based on the authors experience in working with organic, integrated, and conventional food and farming systems.

### 2.1. Pesticide Use Data

Pesticide use on conventionally managed farms at the state and national levels can be tracked through annual USDA pesticide use surveys that were carried out by the National Agricultural Statistics Service (NASS) [87]. The Pesticide Use Data System (PUDS) draws on these USDA data and provides access to detailed information on pesticide use by crop and year, at the national and state level [88]. Major row crops are surveyed in most years, while fruit and vegetable crops are surveyed bi-annually (fruit crops in odd years, vegetables in even years). The data are now accessible via the online, USDA Quick Stats Database [89], and they were previously released in hard-copy reports and electronic data files.

In each year for all crops surveyed, NASS collects data in states that collectively account for at least 85% of the total acres planted nationally to a given crop. The absence of annual survey data for many crops creates gaps in the dataset. To overcome this shortcoming, missing data values are approximated in PUDS through the same technique that is used by the USDA's Economic Research Service to produce continuous use datasets from periodic NASS survey data. The values for years not surveyed are interpolated between years with reported values based on the assumption that pesticide use changes in equal increments year-to-year between two years with reported values. Access PUDS, further documentation, and interactive PUDS tables at: hygeia-analytics.com/puds-the-pesticide-use-data-system/.

The US Geological Survey (USGS) uses survey data and a variety of methods to estimate pesticide use by crop and chemical, drawing predominantly from USDA-NASS datasets [83]. The USGS dataset also provides use estimates at the county level, the only public source of such data. The USGS system can be utilized to generate a variety of data tables and maps of pesticide use over time.

### 2.2. California Pesticide Use Data

The State of California maintains the most detailed pesticide use data system in the world. The California "Pesticide Use Reporting" (PUR) system compiles data by crop, chemical, and county, down to approximately one square mile units identified via an "MTRS" code (meridian, township, range, and section, as defined in the US Public Land Survey) [90]. The CA Department of Pesticide Regulation (CDPR) maintains Geographic Information Systems (GIS) shapefiles with these data for all counties in California [91].

Details on all agricultural pesticide applications that are made within an MTRS block (typically 640 acres/259 hectares) in a year are recorded via mandatory pesticide use reports submitted to County Agricultural Commissioner offices. These PUR records are then forwarded to CDPR for inclusion in the statewide database [92]. The parameters used from the PUR dataset in this study include crop (designated as "site" in the PUR data), site code, county, grower ID, MTRS, site locator ID (a grower-defined field location), active ingredient (AI) applied, pounds AI applied, area treated in acres, and field size (reported as acres planted). The acres treated that are reported by CDPR may be greater than acres planted for several reasons: 1) a parcel is treated more than once and/or 2) a parcel is treated with a product containing more than one AI, as the use of each AI is a treatment. Acres treated may be less than acres planted if only a portion of the field is treated.

The data were converted to SI units of kilograms (kg) and hectares (ha). The application rates in kg/ha treated were calculated from the raw data, and they should fall in the range of the application rates recommended on the label. The application rates in kg/ha planted were also calculated from the raw data and they provide a measure of total applications to the parcel over the year. The percentage of planted hectares that were treated with a specific chemical or group of chemicals was also calculated from the data.

### 2.3. Comparing Pesticide Dietary Risks

Government pesticide residue testing in food has been underway for approximately three decades. The US Department of Agriculture's (USDA's) Pesticide Data Program (US-PDP) and the testing overseen by the Pesticide Residues Committee convened by the United Kingdom's Food Standards Agency (UK-FSA) are the two most extensive food residue monitoring programs [93,94]. Since the early 1990s the US-PDP has focused on foods that make up a sizable share of the diets of infants and children. Between 12,000 and 25,000 samples of 10 to 20 foods are tested annually. The UK-FSA program samples a much broader diversity of foods, food forms, and beverages, but, on average, fewer samples per food. Since 1999, the program has focused on the residue levels and trends in the approximately 80% of the UK food supply that is imported [95].

The US-PDP and UK-FSA both strive to include a percentage of organic food samples roughly proportional to each organic food's market share in their annual sampling protocol. Despite often falling short of this goal, both programs provide opportunities to compare the frequency of residues and the relative dietary risk levels in organic versus conventionally grown food.

The Dietary Risk Index (DRI) system was used to generate the pesticide-residue based tables and figures in this paper [85,96]. A DRI value is an index that is calculated as the ratio of a pesticide's presence in a single serving of food, divided by the maximum level of the pesticide that can be present in the food without triggering the US EPA's "level of concern" (i.e., the maximum allowed daily chronic exposure level for a pesticide in food).

Acceptable, long-term daily intakes of each food-use pesticide are set by the EPA, and they are called "chronic Reference Doses" (cRfD). In the UK, Europe, and most of the world, similar chronic-exposure thresholds are in place and are generally referred to as Acceptable Daily Intakes (ADIs). Such exposure thresholds are expressed as mg of pesticide per kg of body weight per day (mg/kg body weight/day). When the EPA imposes an added safety factor in response to provisions of the 1996 Food Quality Protection Act (FQPA) [97], a cRfD is converted to a chronic Population Adjusted Dose (cPAD) that is equal to the cRfD divided by the FQPA safety factor).

For food-use pesticides, the EPA's cRfD/cPAD sets the amount of a pesticide that can be ingested in a day consistent with the FQPA's "reasonable certainty of no harm" safety standard. FQPA-driven added safety factors are usually 10-X, but, in a few cases, they have been set at 3-X or 5-X.

A given cRfD and cPAD determines the maximum concentration of pesticide $x$ that can be present in a daily serving or servings of food/beverage $y$, without exposing an individual of known weight to a dose of the pesticide that exceeds his or her personal cRfD/cPAD. The Dietary Risk Index system calculates the relative risk levels based on the residues that are detected in a single serving of given food/beverage, coupled with a pesticide's cRfD/cPAD for a person of known weight.

The DRI system uses residue data from the US Pesticide Data Program and the UK-FSA residue testing program to calculate, for a given food–pesticide combination, the mean residue levels across all samples with a reported residue (mean of the positives; %P). The mean residue levels (Mean Res) are coupled with the weight of a single serving of food (Serv) and a person's body weight (BW) to calculate three DRI values:

$$\text{DRI-M} \quad \text{Positive Sample Mean DRI} = (\text{Mean Res} \times \text{Serv})/(\text{cRfD} \times \text{BW})$$

$$\text{FS-DRI} \quad \text{Food Supply DRI} = (\text{Mean Res} \times \text{Serv})/(\text{cRfD} \times \text{BW}) \times \%P = \text{DRI-M} \times \%P$$

$$\text{Sample DRI Individual Sample DRI} = (\text{Pesticide concentration} \times \text{Serv})/(\text{cRfD} \times \text{BW})$$

In these equations, cRfD may be replaced by a cPAD or other ADI, as appropriate. DRI values are dimensionless because they are ratios of two pesticide weights. The DRI can track residues and risk levels by food, by pesticide, in organic versus conventional production, in domestically grown food versus imports, and combinations of the above selection criteria.

The three different ways that DRI values can be computed for any given food–pesticide combination serve different purposes (see Figure 2). The DRI-Mean is appropriate for comparing acute risks across food–pesticide combinations when it is known that residues are present in the food. The Food Supply-DRI is appropriate for the chronic assessment of pesticide dietary risks because it accounts for both mean residue levels and the frequency of residues. Individual sample DRI values are useful in assessing the distribution of residue and risk levels across all the positive samples of a given food–pesticide combination, as well as in identifying the pesticides that account for most of the risk in individual samples, where the foods were produced, and whether they were grown under conventional or organic management.

## Three Ways to Calculate DRI Values

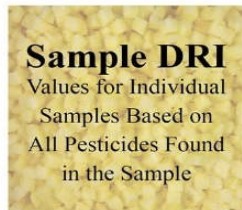 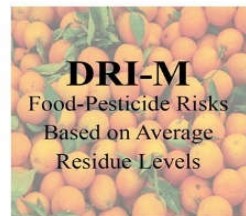 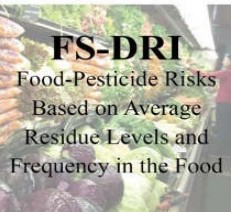

**Figure 2.** A summary of the three ways to calculate Dietary Risk Index values.

For a given food, the aggregate DRI-Mean and FS-DRI values can be calculated by adding the index values across all pesticides that are found in the food. Benbrook and Davis describe the alternative forms and appropriate uses and interpretations of DRI values in detail [85]. Access to the DRI system, methodological details, and interactive tables can be found at hygeia-analytics.com/dietary-risk-index/.

*2.4. Comparing Pesticide Use on Organic and Conventional Farms*

We analyzed the CA PUR data and the use of NOP allowed pesticides to classify specific fields as "organic" or "conventional" by creating a grower and field-specific "Location ID", which is the concatenation of four PUR system data elements: MTRS & Grower ID & Site_Location_ID & Site_Code. Adjuvants and any records lacking an entry in the Grower ID or MTRS field were excluded from the data set, as well as applications of sulfur, horticultural oils, kaolin clay, and insect pheromones, based on US EPA's waiver of data requirements for these substances due to the low toxicity and no requirement for a tolerance on produce grown using these pesticides. The high-volume pesticides sulfur, kaolin clay, and horticultural oils work through non-toxic modes of action by suffocating insects or by creating a barrier between plant leaf and fruit tissues plant pathogens.

We identified organic locations by searching the PUR dataset for active ingredients approved for use in organic production by the NOP (USDA/AMS/NOP 2016) using the dataset of pesticides that are allowed for use on organic farms depicted in Supplemental Table S1. The locations were defined as Organic if ≥98% of hectares that were treated at the location were only treated with NOP-approved pesticide active ingredients. Locations with <98% of hectares treated at the location with NOP-approved pesticides were defined as Conventional.

This approach allowed us to compare pesticide use on a given crop under conventional management as compared to a nearby field producing the same crop, but with pesticide use patterns consistent with organic management and NOP requirements. Using the 2016 CA PUR data, we evaluated three representative crops to compare pesticide use on organic vs. conventional fields, as defined by the Location ID: tomatoes in Yolo County, carrots in Kern County, and grapes in Fresno County. We selected a single county for each crop to ensure that organic vs. conventional locations faced similar weather conditions and regional pest pressures.

The distinction between *hectares treated* and *hectares planted* is an important one for understanding the data. Hectares treated are the actual number of hectares to which the pesticide was reportedly applied during a given application; hectares planted is the size of the field. These numbers are generally not the same for several reasons: (1) growers may only treat a part of the field; and (2) over a season, the field may be treated more than once, resulting in more hectares treated than hectares planted. Note that treatment with a product containing two AIs will be counted as two treatments.

Using the PUR data to determine hectares planted is not always straightforward because of errors in the data set, as pesticide users often confuse the two during data entry. We found the maximum and minimum values of the "acres planted" field in the raw PUR data set for each location and determined the difference between the two to determine *hectares planted* of conventional and organic crops. For each location for which there were differences, we evaluated each reported pesticide use record and used the most common value (the mode) of acres planted.

We compared the organic and conventional locations by kg of pesticides used, hectares treated, application rates in kg/hectare treated and in kg/ha planted, and the percent of planted hectares that are treated both by use type—insecticide, herbicide, fungicide, microbiocide, plant growth regulator, insect growth regulator, or fumigant—and by individual chemical. The Supplemental Material presents data for the individual chemicals.

## 3. Results

*3.1. Intensity of Pesticide Use on Conventional Crops*

Accurate, detailed, and up-to-date pesticide use data are not available from any single public source in the US. The USDA surveys pesticide use on many crops, but not all, and it does not survey all acres of any crop [87]. The EPA periodically releases an agrichemical use report with comprehensive data on total pesticide use on farms and ranches, excluding sulfur and oils. The most recent EPA report provides data for 2012 [98].

Table 1 draws on USDA and EPA data sources to produce estimates of the total pesticide use in US agriculture from 1992 through 2019. The data for 1992–2012 are predominantly from the EPA's periodic pesticide use reports [98–104]. Estimates of total agricultural use in 2019 are from NASS reports and other data sources are explained in Supplemental Table S2. The Supplemental Table File also includes versions of metric-unit tables (kgs and hectares) in English units (pounds and acres). The kilograms of seed treatments and *Bt* toxins that are expressed in plant-protected corn and cotton are estimates based on the acreage planted to various *Bt* crops and average *Bt* endotoxin expression levels (see Supplemental Table S2 for details).

**Table 1.** Estimated kilograms of pesticides applied in US agriculture (million kilograms active ingredient, see notes).

|  | **1992** | **2001** | **2012** | **2019** |
|---|---|---|---|---|
| Herbicides/PGR | 232 | 197 | 256 | 343 |
| Insecticides | 41 | 33 | 15 | 30 |
| *Bt* toxins | 0 | 5 | 64 | 100 |
| Seed Treatments | <1 | 1 | 2 | 2 |
| Total Insecticides | 42 | 39 | 81 | 132 |
|  |  |  |  |  |
| Fungicides | 34 | 19 | 24 | 23 |
| Seed treatments | <1 | <1 | 1 | 1 |
| Total Fungicides | 34 | 20 | 25 | 24 |
| Fumigants | 41 | 46 | 50 | 45 |
| Other | 4 | 5 | 7 | 9 |
| Sulfur/lime, oils, kaolin clay | 30 | 82 | 55 | 59 |
| Total | 382 | 387 | 474 | 612 |
| Cropland Hectares (million) | 132 | 132 | 132 | 132 |
| Avg. Kilograms Pesticides per Hectare | 2.9 | 2.9 | 3.6 | 4.7 |

Notes: 1992–2012 data are from periodic US EPA "Pesticide Industry Sales and Usage" reports. Data for 2019 is from USDA surveys. See Supplemental Table S2 for details regarding how EPA and USDA data were drawn upon in constructing this table and for the basis for estimating *Bt* toxin and seed treatment use.

Across the approximately 132 million hectares (325 million acres) of conventional crops grown in the US in 2019, an average of about 4.7 kg/ha (4.2 lb/A) of pesticide active ingredient was applied on conventionally managed farms. Herbicides accounted for nearly one-half of total pesticide use in 201, and the volume applied has increased 34% since 2012. The spread of resistant weeds on fields planted to GE corn, cotton, and soybeans has accounted for most of the increase.

Insecticides account for the largest increase in mass applied on a percentage basis from 1992 to 2019, increasing 300%. The decline in conventional insecticide use from 1992 through 2012 was brought about by two factors: (1) the shift from relatively high-dose products to lower-dose families of chemistry and (2) the introduction in 1996 and widespread adoption of *Bt* corn and cotton. Table 1 includes estimates of the kilograms of *Bt* toxins expressed in *Bt* corn and cotton. The total kilograms have increased through the broader incorporation of multiple genes expressing different *Bt* endotoxins (e.g., SmartStax corn expresses six *Bt* endotoxins and an estimated total of ~1.9 kg (4.1 pounds) of *Bt* based on typical seeding rates.

Fungicide use has remained relatively stable on most crops (but not corn). The trend toward lower-dose fungicides has increased the total acre-treatments, but not as markedly as in the case of herbicides and insecticides. Fumigant use has fallen in recent years as a

result of regulatory restrictions and voluntary cancellations. In the seven years between 2012 and 2019, the total pesticide use rose 138 kg (304 million pounds), a 29% increase.

The pesticides used on conventional farms is a function of crop grown, farming systems, pest pressure, and the number of preventive practices that are embedded in a farm's IPM system. Such factors vary widely in the US. Table 2 provides a summary of the average pesticide use on a set of conventionally managed crops that account for most agricultural pesticide uses in the US.

**Table 2.** The number of pesticides applied to conventional crops in the US by type of pesticide, year 2019 (see notes).

| | Hectares Planted | Herbicides per Hectare | | Insecticides per Hectare | | Fungicides per Hectare | |
|---|---|---|---|---|---|---|---|
| | (1000 acres) | No. of AIs | Kgs Applied | No. of AIs | Kgs Applied | No. of AIs | Kgs Applied |
| **Row Crops** | | | | | | | |
| Corn | 36,301 | 3.75 | 1.19 | 0.15 | 0.01 | 0.29 | 0.01 |
| Soybeans | 30,797 | 3.43 | 0.97 | 0.21 | 0.01 | 0.32 | 0.02 |
| Cotton | 5559 | 3.60 | 1.52 | 1.36 | 0.27 | 0.04 | 0.00 |
| Sugarbeets | 469 | 4.70 | 0.41 | 0.69 | 0.49 | 1.14 | 0.16 |
| Sunflower | 752 | 1.20 | 0.45 | 0.35 | 0.02 | n/a | n/a |
| Sorghum | 2131 | 3.44 | 1.35 | 0.11 | 0.02 | n/a | n/a |
| Peanuts | 578 | 3.75 | 0.82 | 0.56 | 0.06 | 3.20 | 1.39 |
| Totals | 76,586 | | | | | | |
| **Weighted Average Rate** | | | 1.12 | | 0.03 | | 0.03 |
| | | | | | | | |
| **Grains** | | | | | | | |
| Winter Wheat | 12,610 | 1.69 | 0.29 | 0.08 | 0.00 | 0.35 | 0.02 |
| Oats | 1137 | 0.83 | 0.13 | 0.01 | 0.00 | 0.14 | 0.01 |
| Barley | 1101 | 2.92 | 0.32 | 0.05 | 0.00 | 0.39 | 0.02 |
| Rice | 1028 | 3.29 | 1.37 | 0.29 | 0.01 | 0.84 | 0.05 |
| Totals | 15,876 | | | | | | |
| **Weighted Average Rate** | | | 0.35 | | <0.01 | | 0.02 |
| | | | | | | | |
| **Fruit and Nuts** | | | | | | | |
| Apples | 119 | 1.25 | 0.53 | 4.99 | 1.17 | 4.14 | 4.24 |
| Pears | 18 | 0.90 | 0.73 | 4.75 | 1.07 | 2.67 | 2.60 |
| Fresh Grapes | 49 | 0.99 | 0.54 | 2.28 | 0.78 | 3.84 | 0.94 |
| Peaches | 30 | 1.82 | 1.32 | 3.21 | 1.05 | 3.41 | 3.02 |
| Strawberries | 18 | 0.30 | 0.11 | 5.90 | 2.16 | 5.14 | 7.55 |
| Oranges | 206 | 2.18 | 2.63 | 4.53 | 2.23 | 1.09 | 0.64 |
| Blueberries | 42 | 2.35 | 1.07 | 3.46 | 1.02 | 3.96 | 2.01 |
| Totals | 483 | | | | | | |
| **Weighted Average Rate** | | | 1.52 | | 1.60 | | 2.16 |
| | | | | | | | |
| **Vegetables** | | | | | | | |
| Potatoes | 370 | 2.47 | 1.02 | 2.55 | 0.21 | 5.67 | 3.61 |
| Tomatoes | 114 | 1.54 | 0.84 | 1.30 | 0.40 | 1.10 | 0.99 |
| Lettuce | 106 | 1.05 | 0.51 | 6.12 | 0.41 | 4.09 | 1.34 |
| Green Beans | 114 | 2.34 | 0.89 | 1.04 | 0.09 | n/a | n/a |
| Peas | 54 | 1.75 | 0.54 | 0.30 | 0.03 | 0.14 | 0.01 |
| Spinach | 27 | 0.21 | 0.25 | 0.57 | 0.21 | 0.18 | 1.02 |

| | | | | |
|---|---|---|---|---|
| **Totals** | 786 | | | |
| **Weighted Average Rate** | | 0.85 | 0.23 | 2.06 |

Notes: The above data do not include fumigants, calcium polysulfides, sulfur or copper compounds, kaolin clay, or petroleum oils. Except for instances where NASS reported the pounds applied for these chemicals as undisclosed. In this case, the pounds are accounted for as other fungicides or other insecticides and may be embedded in the above table. This is primarily true for the copper compounds. Herbicides include plant growth regulators and safeners.

The row crops cotton, corn, and soybeans account for the greatest area treated and the highest volumes of pesticide use. Between two and five different herbicide active ingredients have been used on the approximately 73 million hectares (180 million acres) of these three row crops grown in the US in recent years, averaging approximately 1.2 kg of herbicides per hectare.

Table 2 emphasizes the heavy reliance of conventional fruit and vegetable growers on both insecticides and fungicides. Three to five insecticides and two to five fungicides are used on most conventional farms to bring a crop to harvest. In the 1970s–1990s, broad-spectrum insecticides dominated insect pest management, triggering secondary pest outbreaks and the emergence, as well as spread, of resistant insects. In the last 20 years, the industry and conventional farmers have been incrementally moving away from disruptive, broad-spectrum insecticides, and have instead relied more heavily on newer, lower-dose insecticides with modes of action targeting specific physiological, metabolic, or reproductive targets.

This trend has reduced the frequency of secondary pest outbreaks and slowed the spread of resistant pests. Reflecting this trend, the number of different insecticides applied on conventional fruit and vegetable crops has nearly doubled in the last two decades.

Table 2 excludes some of the high-volume pesticides that work through non-toxic modes of action. These include petroleum and horticultural oils and kaolin clay, products that work by creating a barrier between plant leaf and fruit tissues and insects or plant pathogens, as well as lime-sulfur products and copper fungicides that protect crops from plant pathogens via multiple mechanisms. These products are often excluded from pesticide use tables that are issued by the US EPA (e.g., see Tables 3 and 4 in [102]) and USDA reports, because they pose little, if any, known risks, and they are exempt from the requirement for a tolerance. In addition, when they are included in tables, their relatively high rates of application mask significant changes in the use of pesticides that pose risks to humans and the environment at much lower rates of application.

**Table 3.** Use rate and toxicological properties of most used pesticides on certified organic farms compared to widely used conventional pesticides (see notes).

| Organic (o) and Conventional (c) Pesticides | Typical Use Rate [2] (lbs/acre) | Mammalian Toxicity | | | Ecotox | |
|---|---|---|---|---|---|---|
| | | $LD_{50}$ [1] (mg/kg/day) | Chronic RfD (mg/kg/day) | Aggr. DRI-M [2] | Honey Bee Contact $LD_{50}$ (μg/bee) | Daphnia $EC_{50}$ (μg/L) |
| **Insecticides** | | | | | | |
| **Organic** [3] | | | | | | |
| *Bacillus thuringensis*[o] | <0.01 | >20,000 | None | 0 | NA | NA |
| Spinosad[o] | 0.2 | 3738 | 0.02 | 0.0121 | 0.0035 | >1000 |
| Neem oil[o] | 1.1 | >5000 | None | 0 | NA | NA |
| Pheromones[o] | <0.01 | 3250–>15,000 | None | 0 | None | None |
| Pyrethrum[o] | 0.01 | 1400 | None | 0 | | 0.012 |
| **Conventional** | | | | | | |
| Spiroteramat[c] | 0.11 | >2000 | 0.05 | 0.0059 | >100 | >42,700 |
| Imidacloprid[c] | 0.08 | 643 | 0.057 | 0.0345 | 0.0037 | >85,000 |
| Bifenthrin[c] | 0.09 | >2000 | 0.01 | 0.1272 | 0.014 | 1.6 |

| | | | | | | |
|---|---|---|---|---|---|---|
| Methomyl[c] | 1.2 | 10 | 0.025 | 0.2309 | 0.16 | 7.6 |
| Chlorantrani-liprole[c] | 0.075 | >5000 | 1.58 | 0.0023 | >4 | 11.6 |

**Fungicides**

| | | | | | | |
|---|---|---|---|---|---|---|
| Copper (Cupric) ox-ide[o] | 2.85 | >5050 | None | 0 | None | 53 |
| Hydrogen peroxide[o] | 2.6 | 2000 | None | 0 | None | 24 |
| Potassium bicar-bonate[o] | 5.5 | 2064 | None | 0 | >24 | None |
| *Reynoutria* spp (Rega-lia)[o] | <0.01 | >5000 | None | 0 | NA | 50 |
| Boscalid[c] | 0.3 | >5000 | 0.0218 | 0.0084 | 100 | 5330 |
| Chlorothalonil[c] | 0.8 | >10000 | 0.003 | 0.0706 | >40 | 84 |
| Mancozeb[c] | 0.7 | >5000 | 0.05 | n/a | >16 | 13,600 |
| Trifloxystrobin[c] | 0.25 | >5000 | 0.038 | 0.026 | >200 | >95,300 |
| Propiconazole[c] | 0.1 | 1517 | 0.1 | 0.0813 | >25 | 10,200 |

Notes: [1], Rat LD50 unless otherwise specified. Source: US EPA Registration Reviews and European Food Safety Pesticide Peer Reviews, various years. [2]. Typical Use Rates for microbial biopesticides and pheromones active ingredients estimated as <0.01 pound. Aggregate DRI reflects residues in all foods in recent years. [3]. Missing values for organic pesticides result from the granting of a waiver for a data requirement by EPA, low toxicity, and/or low exposure potential. See www.regulations.gov/document/EPA-HQ-OPP-2008-0632-0002, accessed on 22 June 2021.

**Table 4.** Summary of pesticide use on conventional tomatoes in Yolo County, CA in 2016.

| Use Type | kg AI Used [1] | ha Treated [1] | Percent of Planted ha Treated [2] | Ave Rate (kg/ha Treated) | Application Rate (kg/ha Planted) [2] |
|---|---|---|---|---|---|
| Insecticide | 5743 | 36,110 | 283% | 0.16 | 0.45 |
| Herbicide | 35,878 | 31,041 | 243% | 1.16 | 2.81 |
| Fungicide | 14,066 | 25,067 | 196% | 0.56 | 1.10 |
| Insect Growth Regulator | 163 | 750 | 6% | 0.22 | 0.013 |
| Plant Growth Regulator | 472 | 678 | 5% | 0.70 | 0.04 |
| Fumigant | 4833 | 18 | 0.1% | 265.37 | 0.38 |
| Other [3] | 183 | 487 | 4% | 0.38 | 0.014 |
| **Totals** | **61,337** | **94,152** | **737%** | **0.65** | **4.80** |

Notes: Data source is 2016 California Pesticide Use Reporting Data. See Supplemental Table S2 for detailed use by chemical. [1] Excludes the low-toxicity active ingredients sulfur, kaolin, aliphatic petroleum oils, and pheromones, which account for 8% of total hectares treated, 76% of total kg used, and 7% of applications. [2] Total hectares of conventional tomatoes planted in Yolo County by PUR data in 2016 was 12,776. [3] It includes microbiocides, rodenticides, and nematicides.

### 3.2. Pest Management on Organic Farms

Most of the synthetic pesticides are prohibited for use on organic farms. In the United States, the US Department of Agriculture (USDA) established the list of pesticides and other agricultural inputs that may be used on organic farms under the regulatory authority of the National Organic Program (NOP) (see the Introduction and Methods sections for details).

Currently, 44 pesticide active ingredients that are used in organic production are exempt from regulation by the EPA because they pose minimal risk to human health and the environment [19]. The pesticides that qualify for exceptions in the Organic Foods Production Act include copper, horticultural oils, kaolin clay, and sulfur compounds; toxins

that are derived from bacteria; insect traps and sticky barriers; pheromones; and soaps [18].

Supplemental Table S1 includes a full taxonomy of pesticides that are allowed for use on organic farms, organized by the target pest. Table 3, below, provides an overview of the use rates, toxicological, and ecotoxicological metrics for the most widely used pesticides on organic farms. Some of the most widely used synthetic pesticide alternatives are listed below each organically approved insecticide and fungicide. The synthetic pesticide alternatives shown in Table 3 represent the use and toxicity profile of the most heavily applied families of chemistry that are relied upon by conventional growers.

The mammalian toxicity metrics shown in Table 3 are: acute Lethal Dose for 50% of test animals in mice or rats, the EPA-set chronic Reference Dose or chronic Population Adjusted Dose (cPAD) used in conducting dietary risk assessments, and the aggregate DRI mean based on residues in foods that are tested by the USDA's PDP. The higher the DRI-Mean index value, the more worrisome the risk; any values over 0.5 warrant scrutiny to assure compliance with the Food Quality Protection Act's basic "reasonable certainty of no harm" safety standard.

It is noteworthy that the use rates of bioinsecticides are typically lower than the leading synthetic insecticide options, while the opposite is true in the case of fungicides. Chronic toxicity and dietary risks are the largest and most consistent differences between pesticides that are approved for use on organic farms and conventional pesticide alternatives.

Spinosad is widely used on both organic and conventional farms. It is a relatively large and stable molecule that is often detected on fruits and vegetables, especially when applied after fruit or vegetables have begun to form. However, the relatively low residues detected, coupled with spinosad's relatively high cRfD, result in low dietary risks. The dietary risks that are associated with typical spinosad residues in conventional or organic food are 10.5-fold lower than in the case of bifenthrin residues and 19-fold lower than methomyl residues.

There is growing empirical evidence that organic farmers go beyond input substitution for pest management. Certified organic farmers that responded to a 2019 USDA survey reported reliance on many bio-based practices that are known to mitigate or eliminate the need for pesticide applications [105], as shown in Table 4. Maintaining buffer strips to provide habitat for beneficial organism, the use of animal manure to build soil health, and water management practices to prevent spikes in pest populations were three commonly deployed practices. Some organic fruit and vegetable farmers are also successfully incorporating steam and steam plus mustard seed meal fumigation methods into their Organic System Plans [106].

*3.3. Comparison of Pesticide Use Between Conventional and Organic Fields*

We used California Pesticide Use Reporting (PUR) data to compare pesticide use on certified organic farms to nearby conventional farms for three crops: tomatoes, carrots, and grapes. We evaluated the number of hectares treated and the kg of active ingredient applied, application rates in both kg per *hectare treated* and kg per *hectare planted.* We also evaluated the treated hectares as a percent of total hectares planted in an annual crop production cycle. These use-data metrics are reported by type of pesticide (herbicides, insecticides, fungicides, etc.) in the following tables and by specific AI.

3.3.1. Tomatoes

The PUR data indicate that there were 12,780 hectares of conventional tomatoes and 1875 hectares of organic tomatoes planted in Yolo County in 2016, for a total of 14,655 hectares. This compares favorably to the 14,569 hectares that were reported in county-level statistics collected by the State of California [107]. The tomato dataset contained a total of 80 organic fields ("Locations", see Methods) with an average size of 23.4 ha and 475 conventional fields with an average size of 26.9 ha. There are 64 individual growers,

19 of whom managed organic fields. Fifteen of these organic growers (79%) also managed conventional tomato fields.

Pesticides that were applied to conventional tomatoes primarily included insecticides, herbicides, and fungicides, with minor uses of insect growth regulators (IGRs), plant growth regulators (PGRs), microbiocides, nematicides, and rodenticides (Table 4). Insecticides were the primary pesticide applied to organic tomatoes, with minimal use of herbicides, and only minor uses of fungicides, microbiocides, and PGRs. (Table 5). Organic growers did treat their tomatoes with fungicides—but primarily used the low-toxicity fungicide sulfur. For information on the specific AIs used, see Supplemental Table S3 for conventional tomatoes and Supplemental Table S4 for organic tomatoes.

**Table 5.** A summary of pesticide use on organic tomatoes in Yolo County, CA in 2016.

| Use Type | kg AI used | ha Treated | Percent of Planted ha Treated | Ave Rate (kg/ha Treated) | Application Rate (kg/ha Planted) |
|---|---|---|---|---|---|
| Insecticide | 985 | 1752 | 93% | 0.56 | 0.53 |
| Plant Growth Regulator | 21 | 261 | 14% | 0.08 | 0.011 |
| Microbiocide | 24 | 59 | 3% | 0.40 | 0.013 |
| Fungicide | 33 | 36 | 2% | 0.92 | 0.018 |
| Herbicide | 0.3 | 0.4 | 0.02% | 0.87 | 0.0002 |
| **Totals** | **1063** | **2109** | **112%** | **0.50** | **0.57** |

Notes: Data source is 2016 California Pesticide Use Reporting Data. See Supplemental Table S4 for detailed use by chemical. Excludes the low-toxicity active ingredients sulfur, kaolin, aliphatic petroleum oils, and pheromones, which account for 57% of total hectares treated, 99% of total kg used, and 48% of applications.

The application rates of fungicides in kilograms per *hectare treated* averaged 0.56 kg/ha for conventional tomatoes and 0.92 kg/ha on organic tomatoes, reflecting the use of copper oxide and *Burkholderia sp.* with application rates of 2.3 kg/ha and 1.8 kg/ha, respectively, on organic tomatoes when compared to the lower-application-rate fungicides that were used on conventional tomatoes. Most of the synthetic fungicides used on conventional tomatoes have application rates in the range of 0.1–2 kg/ha, with the median application rate in the Yolo County data set of 0.15 kg/ha. The percent of *hectares planted* treated with fungicides in conventional tomatoes was 196%, which meant that, on average, each hectare was treated twice, with an average application rate of 1.10 kg/ha planted. Applications of products with two AIs count as separate applications, so the application of a product with more than one AI counts the hectares treated for each of the AIs. The percent of *hectares planted* treated with non-sulfur fungicides in organic tomatoes was only 2%, with an average application rate of just 0.02 kg/ha across all hectares planted, as seen in Figure 3.

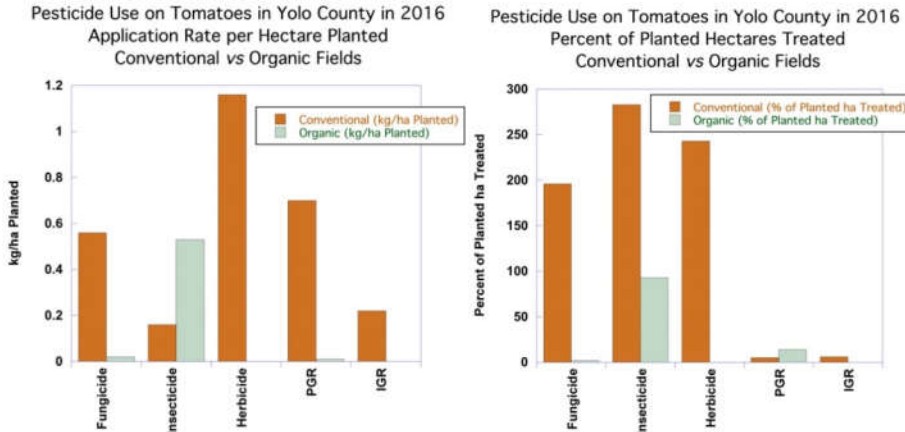

**Figure 3.** Pesticide use on tomatoes in Yolo County in 2016.

The application rates of insecticides averaged 0.16 kg/ha treated for conventional tomatoes and 0.56 kg/ha treated on organic tomatoes, reflecting the somewhat higher PUR-reported application rates of the microbial insecticides *Bacillus thuringiensis (Bt)* and *Chromobacterium subtsugae*. The application rate of the active component of *Bt* in the PUR system is inflated, and it does not reflect kilograms of active ingredient per hectare, as in the case of other pesticides (see Supplemental Table S4 for details). Most synthetic insecticides have application rates that are in the range of 0.01–0.5 kg/ha treated, with a median rate for Yolo County tomatoes of 0.07 kg/ha treated, while the application rate for organic microbial insecticide application rates range from 0.45–0.9 kg/ha, with a median of 0.54 kg/ha treated. The percent of hectares planted treated with insecticides in conventional tomatoes was 283%, which meant that, on average, each hectare was treated almost three times, with an average application rate of 0.45 kg/ha planted. The percent of hectares planted treated with insecticides in organic tomatoes was only 93%, with a comparable application rate to conventional tomatoes of 0.53 kg/ha planted.

The use of herbicides was the most substantial difference between organic and conventional tomatoes. Less than half a hectare (0.02% of ha planted) or organic tomatoes was treated with herbicides, while the 243 percent of planted hectares of conventional tomatoes was treated with herbicides, nearly two and a half treatments per season. Conventional tomato growers used 17 different herbicides. A single organic grower used an herbicide that contained capric and caprylic acid. Weed management in organic tomatoes typically involves crop rotation, black plastic, between row cultivation, hand weeding, and drip irrigation. Transplanted tomatoes are usually less susceptible to heavy weed pressure than tomatoes that are direct-seeded [108].

Fumigants contributed substantially to the kilograms of conventional pesticide use for tomatoes, although only 18 ha were treated. Organic production does not permit the use of synthetic chemical fumigants.

### 3.3.2. Carrots

The PUR data indicate that there were 8680 hectares of conventional carrots and 3150 hectares of organic carrots planted in Kern County in 2016, for a total of 11,830 hectares. This compares favorably to the 12,252 hectares that were reported by county-level statistics collected by USDA [89,109]. The data set contained a total of 99 organic fields (Locations, see Methods) with an average size of 31.8 ha and 313 conventional fields with an average size of 27.7 ha. There are 64 individual growers, four of whom managed organic fields. Two growers managed both organic and conventional fields.

Fungicides, herbicides, and fumigants were used on a substantial fraction of conventional carrots, with minor uses of microbiocides and PGRs (Table 6). Copper-based fungicides were the type of pesticide applied mostly commonly to organic carrots, with minor

use of insecticides and PGRs, and no use of herbicides. (Table 7). For information on the specific AIs used, see Supplemental Table S5 for data on conventional carrots and Supplemental Table S6 for organic carrots.

**Table 6.** A summary of pesticide use on conventional carrots in Kern County, CA in 2016.

| Use Type | kg AI Used [1] | ha Treated [1] | Percent of Planted ha Treated [2] | Ave Rate (kg/ha Treated) | Application Rate (kg/ha Planted) [2] |
|---|---|---|---|---|---|
| Fungicide | 12,112 | 31,714 | 365% | 0.38 | 1.40 |
| Herbicide | 12,464 | 15,815 | 182% | 0.79 | 1.44 |
| Fumigant | 1,400,662 | 5648 | 65% | 247.98 | 161.37 |
| Insecticide | 630 | 2798 | 32% | 0.23 | 0.07 |
| Microbiocide | 170 | 113 | 1.3% | 1.50 | 0.02 |
| Plant Growth Regulator | 0.3 | 68 | 0.8% | 0.005 | 0.00004 |
| **Totals** | **1,426,038** | **56,156** | **647%** | **25.39** | **164.29** |

Notes: Data source is 2016 California Pesticide Use Reporting Data. See Supplemental Table S5 for detailed use by chemical. [1] Excludes the low-toxicity active ingredients sulfur, kaolin, aliphatic petroleum oils, and pheromones, which account for 10% of total hectares treated, 1% of total kg used, and 11% of applications. [2] The total hectares of conventional carrots planted in Kern County by PUR data in 2016 was 8680.

**Table 7.** Summary of pesticide use on organic carrots in Kern County, CA in 2016.

| Use Type | kg AI Used [1] | ha Treated [1] | Percent of Planted ha Treated [2] | Ave Rate (kg/ha Treated) | Application Rate (kg/ha Planted) [2] |
|---|---|---|---|---|---|
| Fungicide | 9772 | 12,983 | 412% | 0.75 | 3.10 |
| Insecticide | 256 | 290 | 9% | 0.88 | 0.08 |
| Plant Growth Regulator | 1 | 225 | 7% | 0.004 | 0.0003 |
| **Totals** | **10,029** | **13,498** | **429%** | **0.74** | **3.18** |

Notes: Data source is 2016 California Pesticide Use Reporting Data. See Supplemental Table S6 for detailed use by chemical. [1] Excludes the low-toxicity active ingredients sulfur, kaolin, aliphatic petroleum oils, and pheromones, which account for 28% of total hectares treated, 83% of total kg used, and 27% of applications. [2] Total hectares of organic carrots planted in Kern County by PUR data in 2016 was 3150.

The application rates of fungicides in kilograms per hectare treated averaged 0.38 kg/ha for conventional carrots and 0.75 kg/ha on organic carrots, reflecting the higher application rates of copper fungicides when compared to synthetics. Most synthetic fungicides used on conventional carrots have application rates in the range of 0.1–2 kg/ha, with the median application rate in the Kern County data set of 0.25 kg/ha. The percent of hectares planted treated with fungicides in conventional carrots was 365%, which meant that, on average, each hectare was treated nearly four times, with an average application rate of 1.40 kg/ha planted. Applications of products with two AIs count as separate applications, so the application of a product with more than one AI counts the hectares treated for each of the AIs. The percentage of hectares planted treated with non-sulfur fungicides in organic carrots was higher than in the case of conventional carrots, at 412%, with an average application rate of 3.10 kg/ha planted, which reflected the use of higher-application-rate copper fungicides, as seen in Figure 4.

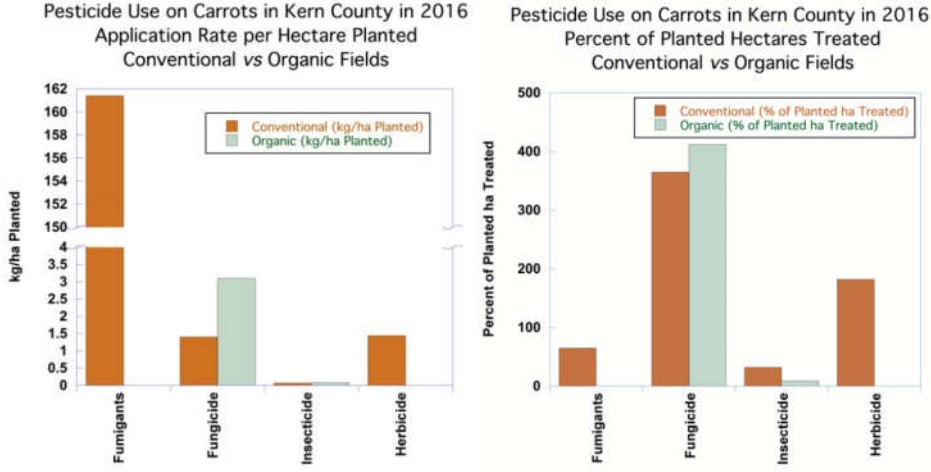

**Figure 4.** Pesticide Use on Carrots in Kern County in 2016.

### 3.3.3. Grapes

The PUR data indicated that there were 45,162 hectares of conventional table and raisin grapes and 3866 hectares of organic table and raisin grapes that were planted in Fresno County in 2016, for a total of 49,028 hectares. The data set contained a total of 292 organic fields with an average size of 13 ha and 3137 conventional fields with an average size of 14 ha. There are 1302 individual growers, 146 of whom manage organic fields. Twenty-seven growers (18%) managed both organic and conventional fields in 2016.

Pesticides that were applied to conventional grapes included primarily fungicides, insecticides, and herbicides, with modest amounts of all other types of pesticides (Table 8). Fungicides, insecticides, and PGRs were the major types of pesticides applied to organic grapes, with only minor use of herbicides and microbiocides, and no use of fumigants, IGRs, bird repellents, and rodenticides (Table 9).

**Table 8.** A summary of pesticide use on conventional grapes in Fresno County, CA in 2016.

| Use Type | kg AI Used [1] | ha Treated [1] | Percent of Planted ha Treated [2] | Ave Rate (kg/ha Treated) | Application Rate (kg/ha Planted) [2] |
|---|---|---|---|---|---|
| Fungicide | 97,813 | 224,343 | 497% | 0.44 | 2.17 |
| Insecticide | 131,723 | 119,626 | 265% | 1.10 | 2.92 |
| Herbicide | 99,242 | 97,195 | 215% | 1.02 | 2.20 |
| Plant Growth Regulator | 11,449 | 44,569 | 99% | 0.26 | 0.25 |
| Insect Growth Regulator | 5780 | 23,406 | 52% | 0.25 | 0.13 |
| Microbiocide | 992 | 686 | 2% | 1.45 | 0.02 |
| Fumigant | 83,559 | 330 | 0.7% | 253.2 | 1.85 |
| Other [3] | 41 | 373 | 0.8% | 0.1 | 0.001 |
| **Totals** | **430,559** | **510,154** | **1130%** | **0.84** | **9.53** |

Notes: Data source is 2016 California Pesticide Use Reporting Data. See Supplemental Table S7 for detailed use by chemical. [1] Excludes the low-toxicity active ingredients sulfur, kaolin, aliphatic petroleum oils, and pheromones, which account for 33% of total hectares treated, 78% of total kg used, and 30% of applications. [2] The total hectares of conventional table grapes planted in Fresno County

by PUR data in 2016 was 45,162.[3] It includes rodenticides, molluscicides, bird repellents, and desiccants.

**Table 9.** A summary of pesticide use on organic grapes in Fresno County, CA in 2016.

| Use Type | kg AI Used [1] | ha Treated [1] | Percent of Planted ha Treated [2] | Ave Rate (kg/ha Treated) | Application Rate (kg/ha Planted) [2] |
|---|---|---|---|---|---|
| Fungicide | 10,206 | 7514 | 194% | 1.36 | 2.64 |
| Insecticide | 2489 | 1917 | 50% | 1.30 | 0.64 |
| Plant Growth Regulator | 117 | 1798 | 47% | 0.07 | 0.03 |
| Herbicide | 429 | 107 | 3% | 4.01 | 0.11 |
| Microbiocide | 46 | 31 | 1% | 1.50 | 0.01 |
| **Totals** | **13,287** | **11,367** | **294%** | **1.17** | **3.44** |

Notes: Data source is 2016 California Pesticide Use Reporting Data. See Supplemental Table S8 for detailed use by chemical. [1] Excludes the low-toxicity active ingredients sulfur, kaolin, aliphatic petroleum oils, and pheromones, which account for 59% of total hectares treated, 95% of total kg used, and 71% of applications. [2] The total hectares of organic table grapes planted in Fresno County by PUR data in 2016 was 3866.

Fungicides were applied to 497% of conventional planted hectares of grapes, indicating that, on average, the planted hectares were treated nearly five times over the season. Applications of products with two AIs count as separate applications, so the application of a product with more than one AI counts the hectares that were treated for each of the AIs. For organic grapes, 194% of planted hectares were treated with fungicides, almost two applications per season. The application rates of non-sulfur fungicides in kilograms applied per ha treated averaged 0.44 kg/ha treated for conventional grapes and 1.36 kg/ha treated on organic grapes, reflecting a greater use of higher-application-rate copper fungicides on organic grapes as compared to the lower-application-rate synthetic pesticides used on conventional grapes (see Figure 5). Forty-seven different fungicides were used on conventional grapes. In contrast, organic growers used 11 different AIs.

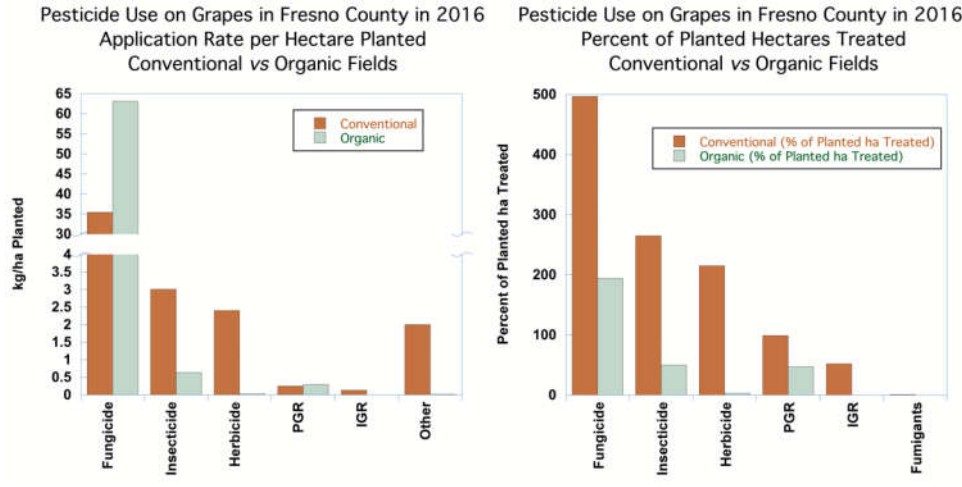

**Figure 5.** The pesticide use on Grapes in Fresno County in 2016.

Insecticides were applied to 265% of the conventional grape acreage and 50% of the organic acreage. The application rates of insecticides in kilograms applied per ha planted

averaged 2.92 kg for conventional grapes and 0.64 kg/ha treated for organic grapes, reflecting much lower use of insecticides overall on organic grapes, coupled with reliance on microbial insecticides that were applied at low application rates.

The use of herbicides was the most significant difference in pesticide use between conventional and organic grapes. For herbicides, 215% of conventional planted hectares of grapes were treated over the season, while only 3% of organic hectares were treated. In organic vineyards, mowing or tilling largely control weeds, while herbicide use dominates weed management strategies in conventional vineyards. The application rates in kilograms per hectare planted for conventional grapes averaged 2.20 kg/ha, while the application rates for organic grapes averaged 0.11 kg/ha planted.

Fumigants contributed substantially to conventional kilograms that were applied for grapes, although only 241 ha were treated, likely for a new vineyard planting.

### 3.3.4. PUR Data Summary

The data shown above and in Table 10 confirm the existence of stark differences between organic and conventional growers in their approach to pest management. Overall, organic growers use low-toxicity pesticides and fewer pesticides per hectare of crop planted and, for the crops evaluated, they are far less dependent on herbicides, insecticides, and fumigants than conventional growers.

**Table 10.** Percent of 2016 applications in CA PUR data consisting of low-toxicity AIs.

| | Organic | | Conventional | |
|---|---|---|---|---|
| | kg AI Used | ha Treated | kg AI Used | ha Treated |
| Tomatoes, Yolo County | 99% | 57% | 76% | 8% |
| Carrots, Kern County | 83% | 28% | 1% | 10% |
| Grapes, Fresno County | 95% | 59% | 78% | 33% |

However, the PUR data presented above obscure the full magnitude of differences in pesticide use on conventional and organic farms. Most of the pesticides applied on organic farms in the above PUR tables pose little or no dietary risks, and generally modest or essentially non-existent ecological risks at the applied rates. The rigorous screening of pesticides by the NOP ensures that organically approved pesticide AIs can be used with minimal risk.

The focus of the analysis in this paper is on higher-toxicity pesticides, and it excludes low-toxicity Ais, including sulfur, horticultural oils, kaolin clay, and insect pheromones, as described in Section 1.3. For the three selected crops evaluated, Table 10 shows the percent of the total kg applied, hectares treated, and the number of applications accounted for by the above listed, very low-risk pesticides. The exclusion of only these AIs still overestimates the risks of pesticide use in organic production, because most of the other NOP-approved pesticides are also low toxicity.

### 3.4. Residues and Risks in Conventional and Organic Food

Table 11 provides an overview of basic residue and risk metrics in the conventional and organic apple samples tested by PDP that year using apples that are grown in the US in 2016 as an example. This is an example of the hundreds of similar tables that are accessible via the DRI system's interactive lookup tables [110] (access at: hygeia-analytics.com/tools/dri/conventional-vs-organic/).

**Table 11.** Residues and DRI risk levels in organic and conventional apples grown in the US in 2016.

|  | Total Samples | Total Number of Residues Found | Percent of Samples with Zero Residues | Ave No. of Residues per Sample | DRI-M | FS-DRI |
|---|---|---|---|---|---|---|
| Conventional | 481 | 2189 | 0.83% | 4.55 | 1.8925 | 0.16 |
| Organic | 16 | 8 | 62.5% | 0.5 | 0.0033 | 0.0004 |
| Ratio of Conventional to Organic | **30** | **274** | **0.013** | **9.10** | **573** | **400** |

Notes: For apples in 2016, this is Table 1 in the interactive look-up tables in the organic versus conventional residue module of the Dietary Risk Index system on Hygeia Analytics [96]. Residue data from the USDA Pesticide Data program.

It is notable that the differences between pesticide dietary risks on organic and conventional apples in 2016 were about five-times greater when based on DRI values, as opposed to the average number of residues in a sample. This is because, in general, the residues of any given pesticide in organic samples are usually markedly lower than the same residue in conventional samples. In addition, the average chronic toxicity of residues in conventional food exceeds the average toxicity of the residues found in organic samples by a wide margin.

Table 12 provides further details of organic apples sampled and tested by PDP in 2016. The total samples tested appears in the first column, followed by the number of residues above the detection limits, the percent of total samples that were positive, the mean residue level, and DRI-Mean and FS-DRI values.

**Table 12.** Pesticide residues detected in organically grown apples, 2016 dietary risk indicators.

| | Total Samples | Number of Positives | Percent Positives | Mean Residue (ppm) | DRI-M | FS-DRI |
|---|---|---|---|---|---|---|
| Diphenylamine (DPA) | 16 | 4 | 0.25 | 0.0075 | 0.00047 | 0.00012 |
| Spinosad A | 16 | 1 | 0.0625 | 0.008 | 0.00199 | 0.00012 |
| Thiabendazole | 16 | 3 | 0.1875 | 0.0043 | 0.00082 | 0.00015 |

Notes: For apples in 2016, this is Table 2 in the interactive look-up tables in the organic versus conventional residue module of the Dietary Risk Index system on Hygeia Analytics [96]. Residue data from the USDA Pesticide Data program.

Table 13 shows how apples in 2016 complied with USDA NOP pesticide provisions. The table identifies whether the residues are from post-harvest (PH) fungicides; the number of positives, the mean of the positives, the EPA tolerance, the 5% threshold of EPA-approved tolerances for pesticides prohibited in organic production and handling, and the number of test results with residues over the organic threshold.

**Table 13.** Residues detected in organic apple samples: compliance with national organic program rule pesticide provisions in 2016.

| Analyte | PH Fungicide | Number of Positives | Organic Mean of Positives (ppm) | Tolerance or Action Level (ppm) | Action Threshold (AT) (5% of Tolerance) |
|---|---|---|---|---|---|
| Diphenylamine (DPA) | Yes | 4 | 0.0075 | 10 | 0.5 |
| Spinosad A | No | 1 | 0.008 | 0.2 | 0.01 |

| Thiabendazole | Yes | 3 | 0.0043 | 5 | 0.25 |
|---|---|---|---|---|---|
| **Totals** | | **8** | | | |

| Analyte | Residues Over Action Threshold | Conventional Mean of Positives (ppm) | Number of Inadvertent Residues | DRI-M |
|---|---|---|---|---|
| Diphenylamine (DPA) | 0 | 0.2874 | 4 | 0.00047 |
| Spinosad A | 0 | 0.005 | 0 | 0.00199 |
| Thiabendazole | 0 | 0.3881 | 3 | 0.00082 |
| **Totals** | | | **7** | **0.00328** |

Notes: For apples in 2016, this is Table 3 in the interactive look-up tables in the organic versus conventional residue module of the Dietary Risk Index system on Hygeia Analytics [96]. Residue data from the US-Pesticide Data Program. "Inadvertent Residues" are those in an organic sample that are one-tenth of less than the mean level of positive samples in the corresponding conventional crop. Organic farmers have little or no control over such inadvertent residues.

Table 13 introduces the concept of "Inadvertent Residues". In the DRI system, a presumed "inadvertent residue" is one that is present in an organic food sample at a concentration that is one-tenth or less of the mean residue level in the corresponding conventional crop. This cut-off is designed to reliably distinguish between a residue in an organic sample that is likely the result of a deliberate illegal application on an organic crop, as opposed to a residue that results from contamination organic farmers cannot control, such as pesticide drift onto an organic field from nearby conventional farms, persistent pesticides that are applied more than three years prior to harvest and taken up by crops from the soil, or pesticide residues that are present in packing, handling, and storage facilities that handle both organic and non-organic products.

The assumption underlying this concept is that it is unlikely that the pesticide was deliberately applied to an organic crop in the hope of controlling a target pest when residues are found in an organic sample at a level one-tenth or less of the mean residue level in the corresponding conventional food. Such a fraudulent application would likely result in a residue level that is close to the mean of what is found in the conventional crop, and rarely would such a residue be lower than 0.1 of the conventional sample mean.

For example, see the data covering the presence of the post-harvest fungicide thiabendazole in organic apple samples in Table 13. One-tenth of the mean of the positives for thiabendazole in conventional apples that year was 0.039 ppm. None of the three organic apple samples that contained a thiabendazole residue in 2016 exceeded this 1/10th of the mean of positives in conventional samples and, hence, all three residues meet the DRI-system's definition of an "Inadvertent Residue".

One organic apple sample in 2016 tested positive for spinosad A with a residue level that was 2.5-times lower than the conventional mean. This result is not surprising given that spinosad formulations are registered and approved for use on both conventional and organic farms, and they are applied at approximately the same rate under both management systems.

Residues and DRI risk levels can be compared in organic versus conventional samples for dozens of fresh fruits and vegetables drawing on data from the US-PDP and/or the UK-FSA. The tables in this section summarize recent residue data and DRI risk levels for five fruits and five vegetables. These foods were selected because of the comparatively large numbers of organic samples that were tested in recent years.

Table 14 summarizes the frequency of residues and DRI risk levels in conventionally grown fruits and vegetables tested by the US-PDP in 2019. Frozen strawberries posed the greatest dietary risk from pesticides by a wide margin, with over 10 residues in or on an average sample, and an aggregate DRI-Mean level of 6.7. This level is clearly well beyond

the EPA's level of concern even based on an equal distribution of DRI-Mean values across the 10 residues detected—approximately 0.67 per residue. The FS-DRI value of 0.17 is also of concern, since there will be many individual samples with the residue levels posing risks well above EPA's level of concern, along with many residues below the EPA's dietary risk threshold.

**Table 14.** Average residues per sample and aggregate DRI values for foods tested by the US-PDP: domestic, conventional Samples in 2019 (no banned OC's, rule of 10 applied).

| | **Average Residues per Sample** | **DRI-Mean** | **FS-DRI** | **Share of All Food FS-DRI** |
|---|---|---|---|---|
| **Fruits** | | | | |
| Strawberries, Frozen | 10.30 | 6.7296 | 0.16891 | 1.4% |
| Tangerines | 2.28 | 0.1347 | 0.05901 | 0.5% |
| Kiwi Fruit | 0.47 | 0.5440 | 0.04016 | 0.3% |
| Cantaloupe | 1.16 | 0.4028 | 0.01790 | 0.1% |
| Orange Juice | 1.45 | 0.0325 | 0.00693 | 0.1% |
| **Total Fruits** | **3.15** | **7.844** | **0.2929** | |
| **Vegetables** | | | | |
| Spinach, Frozen | 6.28 | 3.8391 | 0.38487 | 3.2% |
| Sweet Bell Peppers | 2.86 | 2.9658 | 0.27307 | 2.3% |
| Spinach, Canned | 3.51 | 1.7991 | 0.19391 | 1.6% |
| Hot Peppers | 3.61 | 3.1328 | 0.17552 | 1.5% |
| Mustard Greens | 5.54 | 12.9005 | 0.17097 | 1.4% |
| Greens, Collard | 5.22 | 4.5100 | 0.11908 | 1.0% |
| Basil | 6.78 | 0.6450 | 0.03905 | 0.3% |
| Radishes | 0.66 | 0.4716 | 0.03430 | 0.3% |
| Asparagus | 0.31 | 0.2144 | 0.01733 | 0.1% |
| Cilantro | 6.56 | 0.2602 | 0.00743 | 0.1% |
| Sweet Peas, Frozen | 0.39 | 0.0585 | 0.00483 | 0.04% |
| Tomato Paste | 3.80 | 0.0219 | 0.00211 | 0.02% |
| Cauliflower | 0.99 | 0.0877 | 0.00132 | 0.011% |
| Cabbage | 0.36 | 0.0565 | 0.00101 | 0.01% |
| Sweet Peas, Canned | 0.17 | 0.0063 | 0.00019 | 0.002% |
| Garbanzo Beans, Dried | 0.28 | 0.0072 | 0.00005 | 0.0004% |
| **Total Vegetables** | **2.61** | **30.977** | **1.4250** | |
| **All Foods** | | **131.80** | **12.08** | |
| **Selected Fruit + Vegetables as Percent of All Foods** | | **29.45%** | **14.22%** | |

Notes: Total DRI for all foods includes fruits and vegetables that were not tested by PDP in the year of 2019. These foods are extrapolated from previous years.

The vegetables tested by the PDP in 2019 pose much more serious risks when compared to fruits. Six vegetables have aggregate DRI-M values over 1.0, and six have aggregate FS-DRI values over 0.1. Residues that were detected by PDP in all fruits and vegetables tested in 2019 accounted for over 95% of the sum of aggregate DRI-Mean and FS-DRI values across all crops tested that year. The PDP testing results analyzed via the DRI system show that, in most years, all fruits and vegetables account for over 95% of total pesticide dietary risk across all foods and, in some years, over 98%.

Every year when the US-PDP and UK-FSA test results are released, some conventionally grown fruits and vegetables pose clearly worrisome risks that are based on DRI-Mean and FS-DRI values. These dietary risk metrics are based on the mean residue levels.

Table 15 provides an example of the distribution of residues and risk levels in conventional apples that were tested in 2016. Note that the aggregate DRI level in the highest risk sample is 18-times the DRI associated with the sample at the 50th percentile in the distribution. When assessing the distribution of residue levels of a single pesticide in a crop like apples, the residue level at the 95th of the distribution is generally 2 to 3.5 times higher than the mean residue and, at the 90th percentile of the distribution, the risk levels are generally 2–3 times higher than at the mean.

**Table 15.** Selected individual sample DRI Values for US-grown, conventional apples tested by the US-PDP in 2016.

| Sample ID | Number of Positives | Aggregate Sample DRI | Ranking | Percentile |
|-----------|--------------------|--------------------|---------|------------|
| 211 | 7 | 2.03 | 1 | 100th/Max |
| 313 | 6 | 2.01 | 2 | 99.5th |
| 243 | 6 | 0.208 | 120 | 75th |
| 600 | 2 | 0.112 | 240 | 50th |
| 705 | 3 | 0.0448 | 360 | 25th |
| 521 | 2 | 0.000423 | 475 | 1st |
| 400 | 1 | 0.000313 | 479 | 0th/Min |
| **Total Positive Samples:** | **479** | | | |

Notes: Table generated by the Dietary Risk Index system [96]. Residue data from the US Pesticide Data Program (PDP).

The upper-end residues in the distribution are bound to pose DRI values well above 1 when the DRI-Mean value is 0.5 or above for a given pesticide-food combination, which triggers the EPA's "level of concern". A DRI value of 1 for $pesticide_x$ in $food_y$ means that a person would ingest the maximum amount of $pesticide_x$ allowed by EPA from just one serving of $food_y$ in a single day. Such a level of exposure would leave no room in $pesticide_x$'s "risk cup" for additional residues in different foods.

Table 16 provides an overview of the residue frequency and DRI values in the organic food samples that were tested in 2019 by the PDP. Only about one in every three organic fruit samples contained a residue, while the average conventional fruit sample contained 3.15 (data not shown). The residue levels in the organic fruit posed lower risks in general, as evident in the DRI-Mean levels of 0.21 in organic fruit and 7.8 in the conventional fruit samples.

**Table 16.** Average residues per sample and aggregate DRI values for foods tested by the US-PDP: domestic, organic Samples in 2019 (no banned OC's).

| | Average Residues per Sample | DRI- Mean | FS-DRI |
|-|----------------------------|-----------|--------|
| **Fruits** | | | |
| Cantaloupe | 0.13 | 0.1733 | 0.02167 |
| Tangerines | 1.00 | 0.0365 | 0.01825 |
| Strawberries, Frozen | 0.29 | 0.0022 | 0.00063 |
| **Total Fruits** | **0.29** | **0.2121** | **0.04055** |
| **Vegetables** | | | |
| Sweet Bell Peppers | 0.83 | 0.8999 | 0.15033 |

| | | | |
|---|---|---|---|
| Mustard Greens | 0.92 | 1.2445 | 0.02723 |
| Hot Peppers | 0.40 | 0.0669 | 0.01338 |
| Basil | 2.45 | 0.3662 | 0.01197 |
| Spinach, Frozen | 0.75 | 0.0112 | 0.00452 |
| Sweet Peas, Frozen | 0.08 | 0.0146 | 0.00122 |
| Greens, Collard | 0.64 | 0.0129 | 0.00091 |
| Cilantro | 1.90 | 0.0050 | 0.00054 |
| Garbanzo Beans, Dried | 0.30 | 0.0110 | 0.00053 |
| Radishes | 0.06 | 0.0010 | 0.00006 |
| Cauliflower | 0.08 | 0.0007 | 0.00005 |
| Asparagus | 0.14 | 0.0003 | 0.00004 |
| Tomato Paste | 0.08 | 0.0004 | 0.00002 |
| **Total Vegetables** | **0.75** | **2.635** | **0.2108** |

Notes: Table generated by the Dietary Risk Index system [96]. Residue data from the US Pesticide Data Program (PDP).

In the vegetables that were tested by PDP in 2019, the conventional samples contained residues resulting in an aggregate DRI-Mean value that was 12-fold higher than the organic samples. While 75% of the organic vegetable samples had one residue on average, the conventional vegetable samples contained 2.6 residues. The organic sweet bell peppers that were tested by the PDP in 2019 posed the highest FS-DRI of the organic produce tested that year and accounted for over 70% of the total FS-DRI arising from the 13 vegetables with one or more residues. Seven of the 13 organic vegetables contained residues posing aggregate FS-DRI risks that were below 0.001. Organic fruits and vegetables accounted for over 98% of total DRI risk levels across all foods tested, as in the case with the residues detected in conventional foods by PDP in 2019 (data not shown).

Table 17 shows consistent and mostly substantial differences favoring organic foods in terms of the average number of residues detected per sample and the DRI risk levels. Conventional fruits and vegetables contain many more residues per sample than organic samples (range, 2.7 to 17). The DRI-M values averaged 52 times higher in the conventional vegetables and 132 times higher in conventional fruits (range, 1.5 for spinach to 564 for apples), while the FS-DRI conventional-to-organic values averaged 55 times higher in vegetables and 115 times higher in fruits (range, 2.1 for grapes to 406 for apples). The largest absolute reductions in the DRI values occurred for the aggregate DRI-M in kale, which fell from 8.7 to 0.48 and for the FS-DRI in spinach (from 0.76 to 0.33). The large percentage reductions in canned tomato DRIs occurred in the context of low values in both the organic and conventional samples.

**Table 17.** Residues and DRI values for US-grown conventional and organic foods: US-PDP testing in 2014–2017.

| Production System, Food, and Year | Average Number of Total Samples | Average Number of Residues per Sample | Aggregate Food | | Ratio Conventional to Organic Samples | |
|---|---|---|---|---|---|---|
| | | | DRI-M | FS-DRI | DRI-M | FS-DRI |
| **Celery, 2014** | | | | | | |
| Conventional | 588 | 5.00 | 1.06 | 0.153 | 38.7 | 128 |
| Organic | 24 | 0.292 | 0.0275 | 0.00120 | | |
| **Kale, 2017** | | | | | | |

| | | | | | | |
|---|---|---|---|---|---|---|
| Conventional | 568 | 5.10 | 8.73 | 0.275 | 18.2 | 14.1 |
| Organic | 83 | 1.47 | 0.479 | 0.0195 | | |
| **Spinach, 2016** | | | | | | |
| Conventional | 549 | 8.52 | 4.50 | 0.763 | 1.55 | 2.33 |
| Organic | 51 | 3.13 | 2.90 | 0.327 | | |
| **Sweet Potatoes, 2017** | | | | | | |
| Conventional | 665 | 0.761 | 0.524 | 0.0514 | 5.65 | 9.69 |
| Organic | 35 | 0.114 | 0.0929 | 0.00531 | | |
| **Tomatoes, Canned, 2017** | | | | | | |
| Conventional | 476 | 1.26 | 0.124 | 0.00321 | 194 | 122 |
| Organic | 62 | 0.113 | 0.000639 | 0.0000263 | | |
| | | | | **Average Five Vegetables:** | **51.7** | **55.2** |
| **Apples, 2016** | | | | | | |
| Conventional | 481 | 4.55 | 1.85 | 0.160 | 564 | 406 |
| Organic | 16 | 0.500 | 0.00328 | 0.000394 | | |
| **Apple Sauce, 2017** | | | | | | |
| Conventional | 489 | 4.29 | 0.118 | 0.0150 | 3.90 | 6.46 |
| Organic | 41 | 0.634 | 0.0302 | 0.00232 | | |
| **Grapes, 2016** | | | | | | |
| Conventional | 336 | 5.05 | 0.544 | 0.0396 | 1.50 | 2.08 |
| Organic | 19 | 0.526 | 0.362 | 0.0191 | | |
| **Pears, 2016** | | | | | | |
| Conventional | 586 | 4.63 | 1.89 | 0.193 | 75.4 | 45.2 |
| Organic | 18 | 1.17 | 0.0251 | 0.00427 | | |
| **Strawberries, 2015** | | | | | | |
| Conventional | 589 | 7.87 | 2.47 | 0.160 | 13.4 | 115 |
| Organic | 30 | 0.700 | 0.184 | 0.010 | | |
| | | | | **Average Five Fruits:** | **132** | **115** |

Notes: Table generated by the Dietary Risk Index system [96]. Residue data from the US Pesticide Data Program (PDP).

The reductions in residues and DRI values that are shown in Table 17 underestimate the degree to which organic farmers reduce the pesticide risks in the crops they harvest. In several organic crops, most of the residues that were reported in US-PDP testing are post-harvest fungicides picked up by organic crops as they move through processing plants that also handle conventional crops. Organic farmers have no control over these typically inadvertent residues.

Although the significant reductions in pesticide dietary risks that are displayed in Table 17 are encouraging, issues persist in organic food supply chains. The average number of residues in organic spinach is of concern (3.13), as is the fact that 122 out of 160 residues that were found in organic spinach in 2016 were not inadvertent. This finding raises questions regarding compliance with the NOP rules. There were four samples of organic spinach with residues of chlorpyrifos at a mean-of-positives concentration of 0.0115 ppm in 2016. These are noteworthy because only one conventional sample of spinach contained chlorpyrifos at a concentration that was 5.75-times lower.

## 4. Discussion

Differences in pesticide use and risks are among the most important consequences of pest management system choices on organic versus conventional farms. On the farm, IPM systems and outcomes drive or influence the level of expenditures on pest management, including the distribution of expenditures across labor, cultural practices, and chemicals. IPM system performance impacts crop yields as well as quality, profitability, and sustainability. For workers and the environment, the grower's choice of pest management practices affects the potential for toxic exposures and ecosystem disruption. For consumers, the IPM system performance drives reliance on pesticides and influences the number of residues and levels of risk per serving of food.

The applications of pesticides are limited on most organic farms and rare on organic farms producing most row crops, small grains, and forages for livestock. Pesticides are the dominant means of control in conventional farming systems for these crops [11]. Organic farmers only apply pesticides after biological, cultural, and other non-toxic methods fail to bring pest populations below economic threshold levels, but only when a suitable product is approved for use on organic farms (not always the case).

Applications on organic farms tend to be limited only to those portions of the farm where pest population and disease infestations are threatening crop yields and quality. Most conventional farmers spray all cropped areas with herbicides. Insecticide and fungicide use varies in response to pest pressure, but USDA and the EPA both show that most conventional crops are treated with at least one insecticide, and many are treated with multiple fungicides. In general, regardless of farm management system, pesticide use is generally lower in hot, dry areas where irrigation is typically required, and higher in rainfed cropping regions, especially those with long seasons and hot humid conditions.

Our findings are consistent with previously cited studies reporting that organic farmers use markedly fewer pesticides, and they apply them less often and to limited acreage. The pesticides that are used by organic farmers are, in almost all cases, significantly lower risk than the pesticides applied on nearby, conventionally managed farms growing the same crop. Herbicide use is strikingly low in organic farming systems when compared to conventional, and fumigant use is non-existent.

Heavy reliance on pesticides on conventional farms commonly leads to the emergence and spread of resistant weeds, insects, and plant pathogens. Resistant pest phenotypes sometimes lead to new or more serious human-health risks:

- The widespread use of azole fungicides is apparently undermining the efficacy of drugs that are used to treat *Aspergillus* lung infections in humans [111].
- An excessive reliance on glyphosate-based herbicides since the mid-1990s has triggered the emergence and spread of over a dozen glyphosate-resistant weeds, forcing farmers to spray additional herbicides, more often, at higher rates, including high-risk herbicides that are known to pose cancer, reproductive, and developmental risks [112,113].

Past USDA surveys provide some insights into the practices that are adopted by organic farmers that directly or indirectly impact pest pressure and the performance of bio-intensive IPM (bioIPM) systems [114]. Such systems integrate multiple practices that are designed to suppress pest populations and avoid their spread, coupled with use of biopesticides when pest populations exceed economic thresholds. A significant number of certified organic farmers actively manage habitat for beneficial organisms, utilize biological controls, and select pest- and disease-resistant crop varieties, as shown in Table 18.

**Table 18.** Typical pest-management related production practices on organic and conventional farms (see notes).

| Practice | Estimated Adoption by Production System | |
|---|---|---|
| | Organic | Conventional |
| Cultural Practices | | |

| | | |
|---|---|---|
| Planted cover crops | 30 | 8 |
| Used no-till or minimum till | 36 | 14 |
| Maintained buffer strips | 66 | NR |
| Used water management practices | 48 | 21 |
| Practiced rotational grazing | 29 | NR |
| Planned plantings to avoid cross-contamination | 24 | 17 |
| Alter System Biology | | |
| Maintained beneficial insect/vertebrate habitat | 29 | 4 |
| Released beneficial organisms | 15 | 10 |
| Selected planting locations to avoid pests | 30 | 20 |
| Chose pest resistant varieties | 29 | 8 |
| Fertility and Soil Health | | |
| Produced or used organic mulch/compost | 35 | 2 |
| Used animal manures | 56 | 15 |
| Planted green manures | 46 | 18 |

Notes: "NR" is not reported. Data on adoption of practices by organic farmers is from a 2019 USDA survey [105] encompassing 16,476 growers. Data on adoption on conventional farms is from the 2017 USDA surveys of production practices and the 2000 survey on IPM system tools and tactics [109,115].

The most recent nationwide USDA survey on pest management practices was conducted for the crop year 2000 [115]. Six commodities and two systems aggregated the survey results: fruit and nut farmers and vegetable farmers. In addition, some IPM-relevant questions were included in the 2017 USDA Census of Agriculture [105].

Overall, the survey results show that organic farmers are more likely to release beneficial insects and maintain beneficial habitat. Organic farmers are nearly three times more likely to plant cover crops and grow green manures. The data regarding the planting of pest-resistant varieties are difficult to interpret, because, in the case of the 2000 survey, the results were specifically keyed to "biotech varieties resistant to crops". Organic farmers responding in 2019 were also more likely to use no-till or minimum till than all farmers responding in 2001.

Unlike conventional farms, pesticides cannot be, and never are, the only tool used on organic farms to prevent crop losses. The need for pesticide interventions on both organic and conventional farms can be reduced by the systematic integration of practices, including crop rotations, selection of resistant varieties, cultivation, the development of biologically active soils, plant nutrition—especially avoiding spikes in available nitrogen, releases of beneficial organisms, timing of planting and harvest, and habitat management.

Organic farmers are far less reliant on pesticides, because most of the above practices are essential for profitable organic production yet are rarely systematically deployed on conventional farms. On most conventional farms, pest management entails tracking pest presence and populations and choosing which pesticide to apply, when, and how. The need to apply a pesticide on conventional farms is typically followed by consideration of which product to apply next to avoid the emergence of resistant pest phenotypes, rather than options to change farming systems in ways that prevent pests from reaching economically damaging levels.

Organic farmers have often been among the first growers adopting and perfecting prevention-based bioIPM systems [114,116]. Worldwide, the transition to certified organic production often begins with farming system diversification and a reduction in the pur-

chased chemical inputs. Further active steps are then layered into farming systems to promote above and below-ground biodiversity. The crux of effective pest management systems on organic farms is prevention via the integration of multiple tools, tactics, and strategies [117–119].

A growing number of conventional farmers growing high-value crops are utilizing biological controls, biopesticides, and other pest management approaches that were pioneered on organic farms. This convergence of IPM systems is expanding the market for low-toxicity biopesticides and reducing pesticide risk levels on many conventionally managed farms [120].

Pesticide input substitution—applying bioinsecticides, such as spinosad or *Bt*, to control insect larvae instead of synthetic pyrethroids, carbamates, or organophosphates—is an intermediate strategy in the transition to sustainable organic production. Organic systems must typically go well beyond input substitution and implement multiple agroecological strategies that reduce pesticide use to be sustainable and profitable [121–123].

Pesticide use and risks are markedly lower on organic farms because organic farmers have worked for decades to design farming systems that take advantage of natural interactions among the organisms sharing an agricultural landscape. A case can be made that reliance on managing ecosystems to prevent pest problems is the single most important distinguishing characteristic of organic farms in contrast to nearby conventionally managed ones. Because growers and scientists worked together decades ago to codify the first principles of organic farming, no or minimal use of pesticides emerged as a critical goal and attribute that resonated with farmers, consumers, and rural neighbors. This remains the case to this day.

*Reducing Pesticide Use and Risks Via Growth in Organic Farming*

Fruit and vegetable crops account for most pesticide use on organic farms, and most of the pesticide residues and risk in food come from conventionally managed fruit and vegetable farms. Reducing the pesticide residues and risk in fresh produce is one of the important societal benefits from the transition to organic, and the benefit of most direct concern for many consumers. Farmers, applicators, and farm workers also face different levels of exposure to pesticides and risks on organic and conventional farms, but there has been a lack of studies on the relative pesticide risks that are associated with different farming systems.

There are approximately 1.6 million hectares (four million acres) of fruits and vegetables grown in the US annually. The transition of these 1.6 million hectares—just over 1.2% of total harvested cropland—could eliminate nearly all pesticide dietary exposure and risk. As the scale of the organic fruit and vegetable industry grows, investments will increase in processing and storage facilities that are 100% dedicated to organic produce. Such new organic supply change infrastructure will eliminate post-harvest fungicide use in packing plants as a source of residues in organic produce.

Most public and private investment in pest management science and technology over the last half century has been dedicated to enhancing the efficacy and cost-effectiveness of pesticide-based systems, while a much smaller share of total pest management R&D has focused on pest prevention via biological control and integrated systems [114,124]. The decrease in public funding and privatization of agricultural research resulted in a focus on proprietary technology [117,125–127]. The emergence of agricultural biotechnology in the early 1990s triggered massive investments in the tools and techniques that are required to create herbicide-tolerant crops, a decision that clearly was designed to enhance the ease of use and effectiveness of herbicide-dependent weed management systems.

The remarkable effectiveness of Roundup Ready crops from 1996 through the mid-2000s undercut promising academic research and farmer-driven innovation in multi-tactic weed management systems in many midwestern states. The desire of other pesticide-seed-biotechnology companies to compete with Roundup Ready technology triggered the redirection of a substantial share of research investment to weed management systems

that were predominantly dependent on herbicides. The emergence and spread of glyphosate-resistant weeds and the industry's response of developing multi-herbicide tolerant cultivars has further reduced investments in, and focus on, non-chemical weed management systems.

The enhancement of soil health and biodiversity via more complex farming systems is the strongest path to more productive and sustainable food production in the US and worldwide. Contemporary pest management systems on most conventional farms systematically undermine both. Problems that arise as a result pose challenges for farmers, scientists, and other stakeholders, and they point to the need to re-examine existing policies, institutions, and economic incentives that shape pest management system evolution.

Recent attention that has been devoted to "regenerative agriculture" among food companies and retailers, farm publications, and in published research is leading some farmers to reconsider the adoption of production methods that not only boost soil health, but also reduce fertilizer, pesticide, and tillage costs, thereby increasing resiliency and profits [128]. Constructive change is possible if farmers—that are already stretched to the limits by factors eroding sustainability—can regain some of their historic profit margin that has been lost to increasing seed, pesticide, labor, and other costs. Organic farmers are leading the way by demonstrating the potential for farming with minimal need to purchase pesticides and fertilizers from off the farm.

## 5. Conclusions

The current trends in agricultural research, capital flows, market power, and political clout are likely to prove as difficult to change as contemporary reliance on pesticides. In the interim, organic farmers will continue to perfect alternative methods to enhance soil health and manage complex, biologically diverse systems. The threat of climate change and growing interest in regenerative agriculture and agriculture's role in capturing carbon in soil may expand the realm of possibility in terms of farmer-driven changes, new investments, policy change, and budgetary priorities. The fact that many of the steps that are needed to mitigate agriculture's contribution to climate change align with those needed to stabilize pest management systems may lead to long overdue changes in the balance of prevention versus pesticide treatments in managing pests.

### 5.1. Pesticide Use

Prevention-based bioIPM employed by well-managed organic farms incorporates a broad array of pest management tactics, tools, and practices that, in most seasons on most fields, collectively prevent pests from reaching economically damaging thresholds. Pesticides play no role, or a minor role, on most land in organic food production. High-value fruit and vegetable crops on organic farms often require some biopesticide use to prevent insect or plant pathogen damage. Less than a dozen pesticides that are approved for use on organic farms work via a toxic mode of action. Of these, only one poses sufficient risk to require EPA-set tolerances (spinosad).

Pesticides bear most of the pest management burden on conventional farms. In some regions via certain routes of exposure, the risks are increasing on both the farm and among rural neighbors, consumers, and ecosystems. Dietary exposures to herbicides that are used as desiccants to accelerate harvest are increasingly common. Herbicide volatilization and drift-induced damage to off-target vegetation have become widespread problems, especially in the Midwest and southeast.

In contrast to the high and increasing reliance on herbicides on over 100 million hectares (250 million acres) of cropland in the US, essentially no herbicide is used on organically managed cropland. Conventional farmers could significantly reduce herbicide use if they adopted preventive Integrated Weed Management systems, such as those practiced on organic farms.

In contrast to the progress that has been made in reducing risks to consumers that arise from insecticide use, dietary risks stemming from fungicide use are increasing. Long-

term trends are driving fungicide use upward: shorter and less diverse crop rotations; tightening quality standards; economic pressure for higher yields; higher seeding rates; climate change; and routine use of fungicides in packing sheds to extend the shelf life of fresh produce.

Risks to ecosystems from the use of systemic, persistent insecticides are increasing, with a 48-fold increase in acute toxicity loading to the environment between 1992 and 2014 [129]. This trend is primarily a result of increases in insecticidal seed treatments on row crops and it contributes not just to the reduction in species diversity and numbers of insects, but also significant adverse impacts on pollinators.

### 5.2. Dietary Exposure to Pesticides

Pesticide dietary risk assessments and analyses focusing on the differences in residue and risk levels in organic versus conventional food can be applied to:

- target future residue testing and human biomonitoring investments,
- enhance the efficacy of organic certification and verification programs and policies,
- prioritize investments in pest management science and bioIPM system innovation and adoption,
- assist the food industry to identify and source food ingredients that pose lower risks from pesticides, and
- identify the need for pesticide regulatory reform and guide rulemaking.

The technology and systems are accessible or within reach over the next decade to support a successful shift to organic management of nearly all acreage growing fruits and vegetables in the US. Such a shift of fruit and vegetable acreage producing organic crops should begin as a matter of priority for food destined for baby food and for consumption by women during pregnancy.

### 5.3. Risks, Costs, and Infrastructure Needs

Many conventional farmers continue to find themselves on a pesticide treadmill, particularly with the increased use of herbicides and fungicides. While some progress has been made with integrated farming practices, sustained successes are notable exceptions. Organic agriculture remains only a small niche, in part because of the lack of research and technology transfer and other support infrastructure.

Three infrastructure and policy issues warrant attention. First, organic producers need better access to packing, processing, and storage facilities that are linked into wholesale and retail supply chains. To be competitive, they require economies of scale that are comparable to those now common along the supply chains moving conventionally grown food to consumers.

The development of appropriate technologies for organic producers is a second related issue. A significant share of organic production is harvested from relatively small fields and managed by farmers working fewer acres than most of their conventional neighbors. A large percentage of farmers that become large-scale organic producers start out on small farms. Such operations have different needs for machinery, implements, and other tools. Organic producers are custom-building their own machinery or looking to European or Asian brands. While small farms are a niche corner of the US farm machinery industry, they are the primary focus of farm equipment manufacturers and innovators in other parts of the world. Old and new technologies—such as steam and biofumigants—offer hope for organic farmers to manage pests by non-toxic modes of action [106].

Public education and access to information regarding the significant health, environmental, animal welfare, farmer, and worker benefits that arise when conventional growers successfully switch to organic farming is the third issue. The growing evidence of health benefits from organic food is particularly compelling for families that have, or plan to have, children [41,130–132]. Given the growing list of reproductive problems stemming from prenatal pesticide exposures, the case will likely grow stronger for new goals among

health-conscious consumers, companies, and countries. Converting all or most US fruit and vegetable production to organic farming or conventional systems that prohibit the use of known, high-risk pesticides would provide US farmers and food companies with a competitive edge in markets and among consumers worldwide that are seeking ways to lessen the risk of adverse pesticide health impacts.

Conventional and organic farming systems appear to be converging on farms producing many high-value horticultural crops, but this is clearly not the case on farms that are producing corn, soybean, cotton, wheat, oats, and many other large-acreage human food crops. Investments in organic farming systems and pest management technology will deliver consumer health and environmental benefits that are directly in step with growth in the supply or organic food, and indirectly via incremental change in pest management systems on conventional farms.

As societies work to mitigate climate change and deal with its impacts on ecosystem dynamics, new pest management challenges are bound to arise. Major change is needed—particularly in weed management systems on conventionally managed, large-acreage commodity crops—to make a meaningful contribution to the stabilization of climate and the resiliency of agricultural production systems.

**Supplementary Materials:** The following are available online at hh-ra.org/wp-content/uploads/2021/05/Final_Org_Pest_SUPP_Tabless-AS-SUBMITTED.xlsx: Supplemental Table S1: Pesticides Permitted for Use in USDA Organic Production; Supplemental Table S2. Details and Sources for Estimated Pounds of Pesticides Applied in U.S. Agriculture (million pounds active ingredient, see notes); Supplemental Table S3. Pesticide Use on Conventional Tomatoes in Kern County, CA in 2016; Supplemental Table S4. Pesticide Use on Organic Tomatoes in Yolo County, CA in 2016; Supplemental Table S5. Pesticide Use on Conventional Carrots in Kern County, CA in 2016; Supplemental Table S6. Pesticide Use on Organic Carrots in Kern County, CA in 2016; Supplemental Table S7. Pesticide Use on Conventional Grapes in Fresno County, CA in 2016; Supplemental Table S8. Pesticide Use on Organic Grapes in Fresno County, CA in 2016.

**Author Contributions:** C.B. conceived the paper, carried out the dietary risk analyses, and secured funding. S.K. conducted the analyses of the California Pesticide Use Reporting system use data. B.B. provided details regarding requirements and impacts of the USDA National Organic Program rule. All authors contributed equally to the writing and revisions of the paper. All authors have read and agreed to the published version of the manuscript.

**Funding:** The funding for C.M.B., B.B., and S.K. required to carry out the analytical work, write, and publish the paper was provided by the Heartland Health Research Alliance (www.hh-ra.org, accessed on 22 June 2021).

**Institutional Review Board Statement:** Not applicable.

**Informed Consent Statement:** Not applicable.

**Data Availability Statement:** All pesticide use data are from the USDA National Agricultural Statistics Service, the US Environmental Protection Agency, or the California Department of Food and Agriculture. Access via references in the paper. The Pesticide Use Data system is accessible at hygeia-analytics.com/puds-the-pesticide-use-data-system/, accessed on 22 June 2021 and the Dietary Risk Index systems is accessible at hygeia-analytics.com/dietary-risk-index/, accessed on 22 June 2021.

**Acknowledgments:** Thanks to Rachel Benbrook for assistance in completing the references and final formatting, and to Karie Knoke for help in compiling the pesticide use and dietary residue and risk tables. The authors also thank the reviewers for their constructive suggestions for improvements in the paper.

**Conflicts of Interest:** C.B. has served as an expert witness in US litigation involving pesticides and health outcomes (e.g., Roundup-non-Hodgkin lymphoma, chlorpyrifos-developmental neurotoxicity, paraquat-Parkinson's disease). C.M.B. is currently involved in a study of herbicide use and adverse birth outcomes in the Midwestern US (www.heartlandstudy.org, accessed on 22 June 2021). S.K. has served as an expert witness in US litigation involving the fate and transport of pesticides in the environment. Pesticide Research Institute was under contract with the USDA-NOP to develop

assessments of materials for potential organic certification from 2012–2015. B.B. is an organic farming and food systems consultant and has worked for many organizations involved in the organic certification process (e.g., the Organic Materials Review Institute, Oregon Tilth Certified Organic, Quality Certification Services, Oregon State University, the IPM Institute of North America, Cornell University).

**Authors' Information:** Charles Benbrook, Executive Director, Heartland Health Research Alliance. Susan Kegley, Principal Scientist and CEO, Pesticide Research Institute, Inc. Brian Baker, Affiliate Faculty, Crop and Soil Science Department and Environmental Science Graduate Program, Oregon State University.

**Abbreviations**

| | |
|---|---|
| 2,4-D | 2,4-dichlorophenoxyacetic acid |
| ADI | acceptable daily intake |
| BW | body weight |
| cADI | chronic acceptable daily intake |
| CDPR | California Department of Pesticide Regulation |
| cRfD | chronic reference dose (EPA) |
| cRfC | chronic reference concentration |
| cPAD | chronic population adjusted dose |
| DRI | dietary risk Index |
| DRI-M | positive-sample mean DRI |
| EC | European Commission |
| EPA | US Environmental Protection Agency |
| EU | European Union |
| EWG | Environmental Working Group |
| FDA | US Food and Drug Administration |
| FS-DRI | food-supply DRI |
| FQPA | Food Quality Protection Act of 1996 |
| GBH | glyphosate-based herbicide |
| ha | hectare |
| IGR | Insect growth regulator |
| kg | kilogram |
| lb | pound, 454 g |
| LOQ | limit of quantitation |
| NHL | non-Hodgkin lymphoma |
| NOP | National Organic Program |
| OC | organochlorine |
| OP | organophosphate |
| oz. | ounce, 28.4 g |
| PGR | plant growth regulator |
| PH | post-harvest |
| ppm | part per million by weight, e.g., mg/kg |
| PUR | Pesticide Use Reporting data set |
| RfD | reference dose |
| RACC | reference amount customarily consumed per eating occasion |
| Serv | serving size |
| tbsp | tablespoon, 14.8 mL |
| tsp | teaspoon, 4.93 mL |
| UK-FSA | United Kingdom's Food Standards Agency |
| USDA | US Department of Agriculture |
| US-PDP | USDA's Pesticide Data Program |

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
