# Peer review of "Organic Farming Lessens Reliance on Pesticides and Promotes Public Health by Lowering Dietary Risks"

_agronomy, doi:10.3390/agronomy11071266_

Round 1
Reviewer 1 Report
The article on Organic Farming Lessens Reliance on Pesticides and Public Health Promotes by Lowering Dietary Risks is an important contribution to improving the state of knowledge in the field of organic farming.
Congratulations to the authors.
Author Response
Revisions Requested: None
Author Response:
Thank you for your kind words.
Reviewer 2 Report
The Article by Charles Benbrook et al.,
“Organic Farming Lessens Reliance on Pesticides and Promotes Public Health by Lowering Dietary Risks”,
escribes how organic agriculture is a production system that relies on prevention, ecological processes, biodiversity, mechanical processes, and natural cycles to control pests and maintain productivity. The results support that organic farms apply pesticides far less intensively than conventional farms. Biopesticides used in organic production, work by a non-toxic mode of action, and pose minimal risks to human health and the environment. As a result, risks from pesticide dietary exposures can be largely eliminated by eating organic food, especially fruits and vegetables.
The author is an expert on the subject, for example:
Benbrook, Charles, "Impacts of Changing Pest Management Systems and Organic Production on Tree Fruit Pesticide Residues and Risk," Acta Horticulturae, 2013.
In addition to the above activities, Benbrook has participated in several lawsuits involving pesticide use and risks, agricultural biotechnology, and food labeling.
It may be published with minor corrections:
Line 8, change: font to number 3 (different typeface)
Line 28 & 29, change the initial capital letter: Integrated Pest Management; Food Quality Protection. Replace with: integrated pest management; food quality protection
Line 181 & 182, change whit: Phytophthora infestans a…….
Line 990, The dot's missing next the number 4 “Discussion”
Line 1158, The dot's missing next the number 5 “Conclusions”
I congratulate on a job well done
Best regards
Author Response
Revisions Requested: Minor edits suggested to correct typos and repair formatting errors.
Author Response:
All recommended edits have been made. Thank you for your kind words.
Reviewer 3 Report
The Authors recommend ways to lower pesticide risks by increased adoption of organic farming practices and highlight options along organic food supply chains to further reduce pesticide use, exposures, and adverse worker and environmental impacts.
The work is interesting and well written. The methodology and results are well elaborated.
The literature, instead, should be improved. In particular, I suggest considering the following works:
https://www.researchgate.net/publication/333154920_An_economic_analysis_of_biogas-biomethane_chain_from_animal_residues_in_Italy
https://www.researchgate.net/publication/340661008_Mediterranean_Diet_Patterns_in_the_Italian_Population_A_Functional_Data_Analysis_of_Google_Trends
I encourage the authors to refine their paper to make it available for publication in the journal.
Author Response
Recommended Revisions: Request to improve literature citations, as well as clarify how the conclusions are supported by the results.
Author response:
The reviewer kindly suggested two specific references to include. However, these references were not appropriate for this manuscript. But we did include several new references in the Conclusion of the manuscript to address the reviewers concerns that the literature cited needed to be enhanced.
More information would be needed from the reviewer to specifically address further concerns with the manuscript.
Thank you for your comments.